# How antisolvent miscibility affects perovskite film wrinkling and photovoltaic properties

Seul-Gi Kim[1], Jeong-Hyeon Kim[1], Philipp Ramming[2,3], Yu Zhong[2,3], Konstantin Schötz[3], Seok Joon Kwon[1,4], Sven Huettner [2], Fabian Panzer [3] & Nam-Gyu Park [1✉]

Charge carriers' density, their lifetime, mobility, and the existence of trap states are strongly affected by the microscopic morphologies of perovskite films, and have a direct influence on the photovoltaic performance. Here, we report on micro-wrinkled perovskite layers to enhance photocarrier transport performances. By utilizing temperature-dependent miscibility of dimethyl sulfoxide with diethyl ether, the geometry of the microscopic wrinkles of the perovskite films are controlled. Wrinkling is pronounced as temperature of diethyl ether ($T_{DE}$) decreases due to the compressive stress relaxation of the thin rigid film-capped viscoelastic layer. Time-correlated single-photon counting reveals longer carrier lifetime at the hill sites than at the valley sites. The wrinkled morphology formed at $T_{DE} = 5\,°C$ shows higher power conversion efficiency (PCE) and better stability than the flat one formed at $T_{DE} = 30\,°C$. Interfacial and additive engineering improve further PCE to 23.02%. This study provides important insight into correlation between lattice strain and carrier properties in perovskite photovoltaics.

[1] School of Chemical Engineering, Sungkyunkwan University (SKKU), Suwon, Korea. [2] Department of Chemistry, University of Bayreuth, Bayreuth, Germany. [3] Chair for Soft Matter Optoelectronics, University of Bayreuth, Bayreuth, Germany. [4] Nanophotonics Research Center, Korea Institute of Science and Technology (KIST), Seoul, Korea. ✉email: npark@skku.edu

Since the pioneering reports on the ~10% efficient solid-state perovskite solar cell (PSC) in 2012[1,2], demonstrating long-term stability by resolving the dissolution issue of organic–inorganic lead halide perovskite in photoelectrocehmical-type solar cell employing liquid electrolyte[3,4], the perovskite photovoltaics has surged swiftly. As a result, the power conversion efficiency (PCE) as high as 25.5% has been achieved in 2020[5]. Although the composition of perovskite started with methy-lammonium lead iodide, abbreviated to MAPbI₃, the recent excellent performing PSCs are based on formamidinium lead iodide, abbreviated to FAPbI₃[6–8] or its derivatives with a certain amount of other cations in FA-site and/or bromide in I site[9–11]. Along with compositional engineering for making progress toward higher PCE, methods for controlling crystal growth have significantly contributed to producing defect-less high-quality perovskite films[12,13]. Forming Lewis acid–base adduct inter-mediate via antisolvent engineering was widely adapted to achieve large perovskite crystals with less grain boundaries[14,15], instead of a direct conversion of wet film to the perovskite phase. Despite the enlarged perovskite grains, the crystal growth by solvent engi-neering can hardly manipulate morphology of perovskite layer. Epitaxial growth, for example, could provide crystal growth nor-mal to the substrate, which is expected to be beneficial to carrier transport. In addition, a flat perovskite surface induced by the solvent engineering may not be effective in optimizing light in-and out-coupling. Thus, it is still required to develop a metho-dology enabling an opto-electronically optimized perovskite layer.

Recently, an approach to control the perovskite morphology has been explored. For example, microscopic wrinkles have been observed for a certain composition of perovskite that suffers buckling of the perovskite thin film[16,17]. In particular, the buckling was explained as a result of local compressive stress relaxation[17]. However, detailed and comprehensive studies for effects of the microscopic wrinkles on the photovoltaic perfor-mances, as well as wrinkling mechanism have not been reported yet. Here, we report a simple and yet effective experimental approach to control and optimize the microscopic geometry of the wrinkles of perovskite thin films to maximize the photovoltaic performances, as well as long-time durability. We also suggest a theoretical model elucidating the wrinkling mechanism based on the detailed experimental data. To control the wrinkled mor-phology, we have designed an experimental method based on temperature-dependent miscibility of dimethyl sulfoxide (DMSO) with diethyl ether (DE) and composition optimization of per-ovskite materials (i.e., $FA_{1-x}MA_xPb(Br_yI_{1-y})_3$, $FA_{1-z}Cs_zPb(Br_yI_{1-y})_3$ and $MA_{1-w}Cs_wPb(Br_yI_{1-y})_3$). To study the detailed mechanism of the wrinkling, we suggest a bilayer wrinkling model with a the-oretical analysis supported by numerical simulations and experimental measurement of optical diffraction. We extend a scope of the study to the investigation of the effects of the microscopic wrinkles on the charge carrier dynamics with time-correlated single-photon counting (TCSPC) coupled with fluor-escence lifetime imaging microscopy (FLIM) and photo-conductive atomic force microscope (pc-AFM). From the combined experimental data, we have found that the wrinkled morphology notably facilitates the charge carriers transport inside the perovskite films.

## Results and discussion
### Formation of wrinkled morphologies depending on perovskite composition.
Experimental procedure to control microscopic wrinkles in PSC is schematically illustrated in Fig. 1a. The per-ovskite precursor solution is first spin-coated on a solid substrate for 20 s, followed by dripping DE 10 s right after spinning. The cross-sectional profile of the microscopic wrinkles can be represented as a sinusoidal curve with a wavelength ($\lambda$) and an amplitude ($A$). $\lambda$ is estimated by calculating the governing char-acteristic periodic length scales from the 2D Fourier transform of optical microscope images (Supplementary Fig. 1) and $A$ by cal-culating the average height difference between the hill ($h_{hill}$) and the valley ($h_{valley}$) such that $A = (h_{hill} - h_{valley})/2$. From the experiments, we found that the wrinkling geometry is affected by substrate temperature ($T_{Sub}$) and the temperature of diethyl ether ($T_{DE}$), as well as the perovskite composition. For example, the wrinkle geometry exhibited a dependence on the compositions of FAPbI₃ perovskite. Hereby MA (or Cs) and Br are used as a substitute for FA and I in FAPbI₃ to form nominal compositions of $FA_{1-x}MA_xPb(Br_yI_{1-y})_3$ and $FA_{1-z}Cs_zPb(Br_yI_{1-y})_3$, and MA and I are partially substituted with Cs and Br in MAPbI₃, leading to $MA_{1-w}Cs_wPb(Br_yI_{1-y})_3$. Figure 1b, c show detailed phase diagram of the wrinkle geometries as a function of the composition para-meters (i.e., $x$ and $y$ for Fig. 1b, and $z$ and $y$ for Fig. 1c, respectively) (see Supplementary Fig. 1a, b for entire experimental data for the wrinkled morphologies). We observed that the wrinkling occurs in selective range of the compositions (i.e., $0 \le x \le 0.4$ and $0.2 \le y \le 0.8$) or ($0.6 \le x \le 0.8$ and $0.4 \le y \le 0.6$) in $FA_{1-x}MA_xPb(Br_yI_{1-y})_3$ and ($z = 0.1$ and $0 \le y \le 0.8$), ($z = 0.2$ and $0 \le y \le 0.6$), ($z = 0.3$ and $0.2 \le y \le 0.4$) or ($z = 0.4$ and $y = 0.2$) in $FA_{1-z}Cs_zPb(Br_yI_{1-y})_3$ Notably, we also observe that there is no wrinkling in MAPbI₃ or its derivatives (see Supplementary Fig. 1c, d). In addition, hundred percent Br ($FA_{1-x}MA_xPbBr_3$) would not lead to wrinkling (i.e., for the $FA_{1-x}MA_xPb(Br_yI_{1-y})_3$ perovskite, $\lambda$ tends to increase with decreasing $x$ (Fig. 1b)). From these experimental observations, it is obvious that the presence of limited amount of Br plays an important role in introducing the wrinkled morphology. With a composition of Br in the range of $0.2 \le y \le 0.8$, $\lambda$ decreases with increasing the composition of MA (i.e., $\lambda$~20 μm for $0 \le x \le 0.2$, $\lambda$~15 μm for $0.2 \le x \le 0.4$, and $\lambda$~8 μm for $x \approx 0.6$). It is also notable that wrinkling hardly occurs for higher composition of $x$ and $y$ such that being >0.8. We also observe that A is relatively shallow (~65 nm) for the composition range with $0 \le x \le 0.4$ and $0.2 \le y \le 0.8$, while is deep (~100 nm) for the composition range with $0.1 \le x \le 0.3$ and $0.3 \le y \le 0.6$. As shown in Fig. 1c, sub-stitution of FA with Cs, the wrinkled morphology is accompanied by further amplified value of $A$ ( > 100 nm) in the range of $0.05 \le z \le 0.4$ and $0.2 \le y \le 0.6$. The sampled composition can be expressed as $(FAPbI_3)_{1-z}(CsPbBr_3)_z$ in case of $z = y$ and more detailed phase diagram and images of the wrinkled morphology are provided in Supplementary Fig. 2a, b, and d. Interestingly, the as-spun films prior to the annealing also show a wrinkled mor-phology in the range of $0.1 \le z \le 0.2$ (Supplementary Fig. 2a), which is sustained even after the annealing (Supplementary Fig. 2b), with slightly decrease in $\lambda$ (Supplementary Fig. 2d). This indicates that the solvent evaporation during the annealing pro-cess hardly affects the wrinkle framework, as is the context in the case of $(FAPbI_3)_{1-x}(MAPbBr_3)_x$ in Supplementary Fig. 2c, e. The annealed films show a decrease in the value of $\lambda$ from 19.7 to 13.9 μm when increasing $z$ from 0.1 to 0.2, while an increase in the value of $A$ from 46.5 nm ($z = 0.1$) to 112.5 nm ($z = 0.2$).

**Effect of $T_{DE}$ on wrinkled morphologies.** Next, with a pre-sentative sample with the maximum amplitude (i.e., $(FAPbI_3)_{0.875}(CsPbBr_3)_{0.125}$ ($z = 0.125$)), we have examined the effects of $T_{Sub}$ and $T_{DE}$ on the wrinkling morphology as shown in Fig. 1d. With fixed $T_{Sub}$, $\lambda$ decreases and $A$ increases with decreasing $T_{DE}$. The effects of $T_{DE}$ on the wrinkled morphology can be explained by the microscopic phase separation of solvent and antisolvent mixture in which the lower the mixture tem-perature the less molar fraction of the solvent, which in turn shorter wavelength and greater amplitude. Detailed theoretical

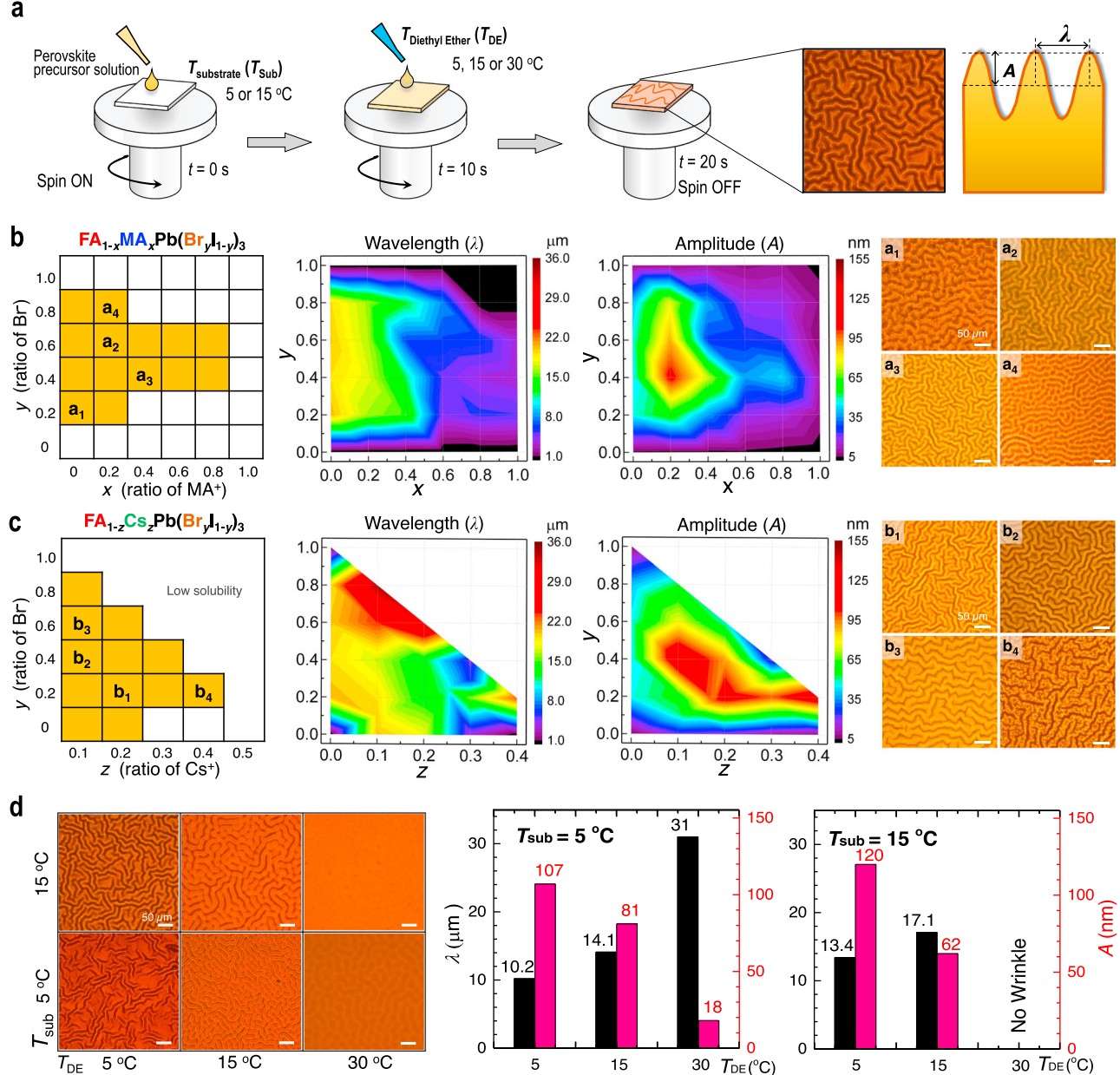

**Fig. 1 Wrinkled morphologies of perovskite thin film. a** A schematic illustration of the experimental procedure to control wrinkled morphology in the perovskite film, together with an optical microscope image of the wrinkled morphology having amplitude ($A$) and wavelength ($\lambda$). Phase diagrams of the wrinkled morphology for (**b**) $FA_{1-x}MA_xPb(Br_yI_{1-y})_3$ (annealed at 145 °C for 10 min) and (**c**) $FA_{1-z}Cs_zPb(Br_yI_{1-y})_3$ (annealed at 145 °C for 10 min) perovskite thin films with different compositions. Color maps of wavelength ($\lambda$) and amplitude ($A$) are presented in the middle panels for each composition. Selected optical microscope images of $a_1$–$a_4$ and $b_1$–$b_4$ are presented in the right panels for each composition. **d** Optical microscope images for the $FA_{0.875}Cs_{0.125}Pb(Br_{0.125}I_{0.875})_3$ perovskite films with different temperatures of $T_{DE}$ and $T_{Sub}$. Scale bar is 50 μm. Dependence of $A$ and $\lambda$ values on $T_{DE}$s at a given $T_{Sub}$ is illustrated with bar graph on the right panels.

analysis on the effects of $T_{DE}$ can be found in Supplementary note 1. In addition, there are no wrinkles at sufficiently high $T_{DE}$ such as 30 °C at a given $T_{Sub} = 15$ °C.

**Bilayer model for wrinkling mechanism.** The formation of wrinkling can be explained with a simple thin film mechanics model. In particular, we can employ a bilayer model in which wrinkles form in the course of relaxation of a compressive stress developed in the perovskite film. It was reported that the compressive stress is due mainly to the differences in the thermal expansion coefficients between the perovskite precursor solution

and the substrate[18]. Previous study on the wrinkling of perovskite thin films suggested that the compressive stress can be developed by the volume change during a fast perovskite formation by using wafer curvature stress measurements[17]. This mechanism requires relatively long-time wrinkle formation dynamics up to several minutes to hours[19,20]. However, we observe that the wrinkles form within 10 s. Therefore, we have developed a more detailed model by which the overall morphology of wrinkles can be elucidated, as well as the wrinkling mechanism based on previous reports[17,18]. Using a model based on the thin film mechanics, the wrinkle geometry can be described as a function of the thickness and mechanical constants of the materials. We also derive

relationships of $\lambda$ and $A$ of the wrinkles with the compositions and $T_{DE}$ (see detailed analysis in Supplementary note 1).

To confirm the wrinkling mechanism, in situ photoluminescence (PL) and simultaneous absorption are examined within the first 24 s of spinning (see ref. [21] for details of the setup). PL and absorption signals commonly emerge immediately after the contact with DE with duration of spinning of 10 s, indicating that perovskite phase forms. As shown in Fig. 2a, the PL peak initially occurs at 712 nm and then shifts to longer wavelengths from 10 s to 24 s. A faster change in PL from 712 nm to over 760 nm is observed for $T_{DE} = 5$ and 15 °C, whereas a more gradual change is observed for $T_{DE} = 30$ °C (Fig. 2b). Extracting the band edge evolution from the in situ absorption data (Fig. 2c), and comparing it to the corresponding evolution of PL peak position show, that early after dripping diethyl ether, the PL peak position is at a lower wavelength than the band edge (Supplementary Fig. 3). We interpret this as the optical signature of a confinement effect, while the subsequent continuous red-shift of PL and band edge indicates a more Cs and Br rich stoichiometry at early times followed by more pronounced incorporation of FA and I at longer times (see Supplementary note 2 for details). We also observe that absorbance becomes stronger at later stage and with increased $T_{DE}$. This indicates that the growth rate of the thin film thickness ($h_f$) is faster for higher $T_{DE}$. The absorbance variation dynamics is translated into the thin film growth dynamics using the absorption coefficient of the perovskite thin films and reference data at equilibrium as shown in Fig. 2d demonstrating that the thickness of the perovskite layer becomes thinner as $T_{DE}$ decreases. (see Supplementary Fig. 3d and Supplementary note 2 for detailed procedure). It is also notable that the PL red-shift is faster for lower $T_{DE}$ (Fig. 2b).

From the in situ PL and absorbance spectra, we can propose that the spin-coated perovskite thin film suffers morphological evolution from viscoelastic layer to elastic layer-capped viscoelastic bilayer. The formation of the elastic capping layer is due mainly to the antisolvent, which drives solvents out of the top part of the coated thin film. In particular, solution of precursor materials of perovskite in DMSO suffers rapid phase separation by introducing the antisolvent DE at the top surface of the film. This leads to the fast crystallization of the top region of the spin-coated layer, which turns into the thin capping layer. The phase separation at lower temperature also gives rise to higher concentration of the perovskite in the top capping layer, which can explain the dependences of the wrinkle geometry on $T_{DE}$ (see Figs. 1d and 2f, and details in Supplementary note 1 with mixing behavior test in Supplementary Fig. 4). In particular, as shown in Supplementary Fig. 4, we observe higher miscibility at relatively higher temperatures (i.e., >25.4 °C), while phase separation of DMSO and DE mixture at relatively low temperature (i.e., <15.2 °C) indicating less miscibility, whereas dimethylformamide (DMF), the main solvent of precursor solution, is fully miscible with DE regardless of the mixing temperature. In contrast, the underlying bottom layer is different from the top capping layer. In particular, as shown in PL and absorbance spectra at earlier stage, the underlying layer exhibited no distinct peaks or band edges signals, and therefore, can be considered as amorphous layer. In addition, the bottom layer is expected to be viscous as demonstrated in the supporting experiment, where DE is poured into the $(FAPbI_3)_{1-z}(CsPbBr_3)_z$ and $(FAPbI_3)_{1-x}(MAPbBr_3)_x$ precursor solution (Supplementary Fig. 5). The viscous precipitate (Supplementary Fig. 5a, b) is maintained for 2–5 min and then converted to the solid phase (Supplementary Fig. 5c, d), while precipitates in $z \geq 0.25$, $x \geq 0.8$ and MAPbI$_3$ are immediately transferred to solid phase with high viscosity ($\eta$) of about $10^{5-7}$ Pa·s (Supplementary Fig. 6). As shown in Fig. 2e for the case of relatively low dynamic viscosity of the viscous precipitate

(samples shown in Supplementary Fig. 5a, b), the viscosity in the bottom layer indicates that the layer is assuredly different from the top capping layer. Based on the in situ absorption and emission studies, combined with dynamic viscosity measurements, we conclude, that the wrinkled structure is likely formed via a bilayer intermediate with a perovskite top layer on a viscous amorphous bottom layer (case 2 in Fig. 2f), while a flat surface results when not undergoing the bilayer intermediate (case 1 in Fig. 2f). Interestingly, too high viscosity (solid precipitate) with the composition of $z \geq 0.3$ also leads to no wrinkle formation as shown in the Case 3 in Fig. 2f. This indicates that the compressive stress of the underlying layer is not dissipated in a local manner, which results in disordered morphological deformation as shown in the optical microscope image in the Case 3. These findings are clearly different from the previous study, suggesting that the crystallization (or nucleation) occurs from the bottom[17].

**Experimental evidence and numerical simulation for bilayer model.** To further confirm the bilayer model for the wrinkling mechanism, we have numerically simulated the morphological evolution of the thin film wrinkling based on temporal evolution of the wrinkle geometry (see details in Supplementary note 3)[22]. As shown in Supplementary Figs. 7 and 8, we can find that the bilayer model provides qualitatively similar wrinkling morphologies accompanied by two-dimensional (2D) fast Fourier transform (FFT) images to the experimentally observed images. We have also tested again the bilayer model by examining the optical diffraction patterns of the wrinkled thin films (Fig. 2g). As shown in Fig. 2h, the optical diffraction patterns would exhibit different patterns (i.e., concentric ring patterns for the wrinkled bilayer, while dot or single ring pattern for the wrinkled monolayer) with different configurations as denoted in Fig. 2g. Indeed, we observe concentric ring patterns at glass side (bottom) of film just after contacted with diethyl ether (10 s after spin started), and the patterns disappears with time, whereas the transmitted concentric ring patterns was sustained for long time as shown in Fig. 2i. This can be compared to the diffraction patterns of the wrinkled perovskite films obtained from reflected side and transmitted side, which are commonly sustained over long time (Supplementary Fig. 9a, b). With the theoretical analysis supported by numerical calculations and experimental observations of the diffraction patterns, we can suggest that the wrinkling of the perovskite thin films can be elucidated by a bilayer model.

**Comparison of photovoltaic property between wrinkled and flat morphology.** To study the effect of bilayer-engineered wrinkled perovskite layer to photovoltaic property, current density($J$)–voltage($V$) characteristics and external quantum efficiency (EQE) of PSCs are measured as shown in Fig. 3a, b. It is evident that the wrinkled morphology affects mainly the open-circuit voltage ($V_{oc}$) and fill factor (FF), while $J_{sc}$ remains nearly unchanged. This is due mainly to a fact that the wrinkled texture is formed at the back-contact side, and therefore, does provide negligible effects on the light absorption or anti-reflection. Indeed, the integrated $J_{sc}$ based on the EQE spectrum in Fig. 3b is calculated to be 22.318 mA/cm$^2$ ($T_{DE} = 30$ °C) and 22.567 mA/cm$^2$ ($T_{DE} = 5$ °C), which is well consistent with the measured $J_{sc}$ values indicating overall change in the light absorption of solar spectrum induced by the wrinkled morphology is nearly negligible. More specifically, the amplitude of the wrinkles (~100 nm) is sufficiently smaller than the quarter of the wavelength of visible and near infrared light. In addition, the spatial periodicity such as wavelength of the wrinkles is sufficiently greater than the wavelength of visible and near infrared incident light, which limits the grating effects of the micro-structures. As listed in Supplementary

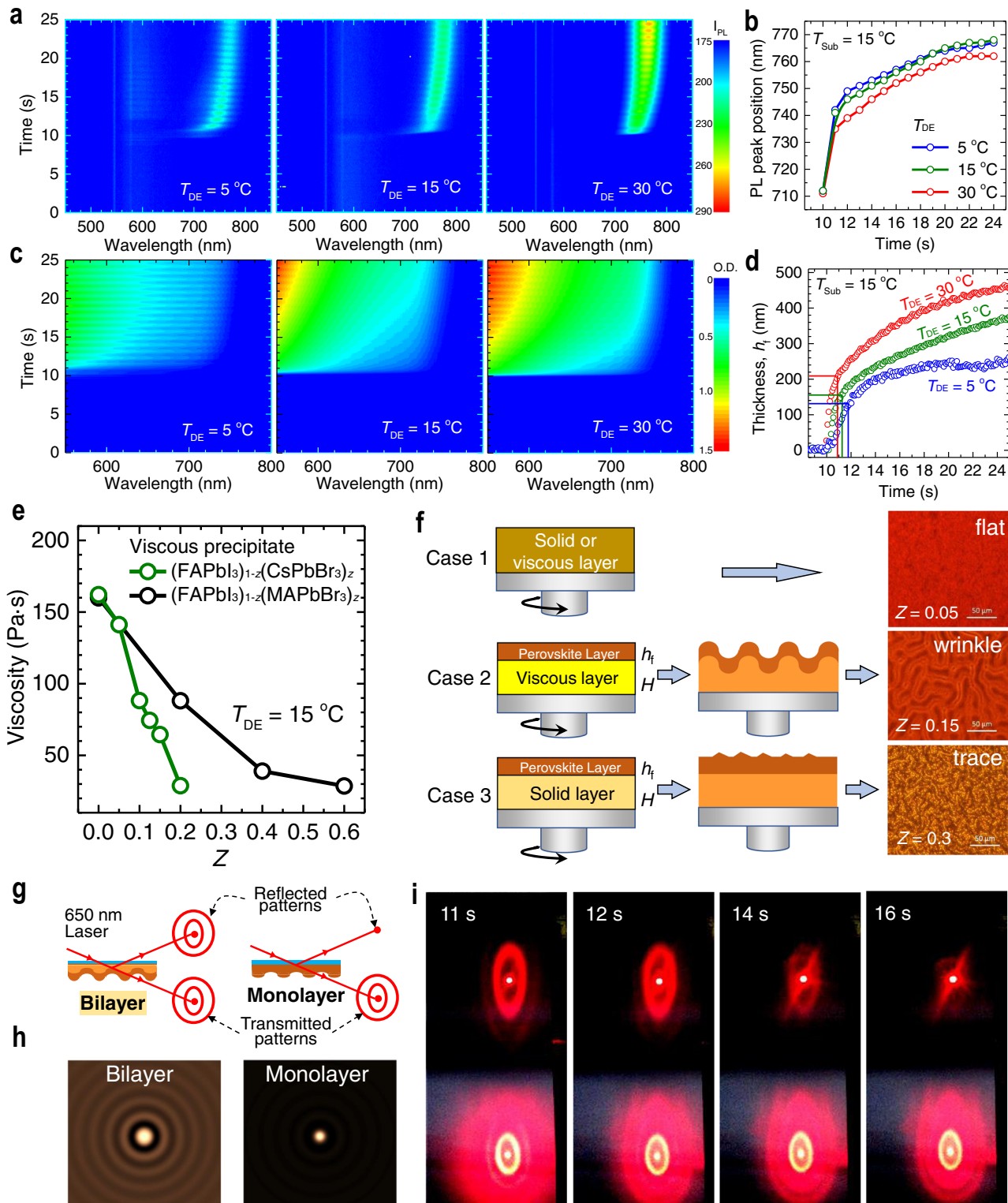

**Fig. 2 Bilayer-induced wrinkling mechanism. a** In situ photoluminescence (PL) measured in the course of spinning from 0 s to 24 s at $T_{Sub} = 15\,°C$ and $T_{DE} = 5$, 15, and 30 °C. **b** Plot of PL peak positions as a function of spinning time. Spin-coating duration was 20 s (dashed line). **c** In situ UV–Vis absorption spectra measured during spinning from 0 s to 24 s at $T_{Sub} = 15\,°C$ and $T_{DE} = 5$, 15 °C and 30 °C. **d** The evolution of the thin solid upper perovskite layer thickness ($h_f$) calculated from the measured absorption data. Spin-coating duration was 20 s. **e** Viscosity of an amorphous viscous precipitates obtained by pouring diethyl ether to the perovskite precursor solution for $(FAPbI_3)_{1-z}(CsPbBr_3)_z$ with $z = 0$–0.2 and $(FAPbI_3)_{1-x}(MAPbBr_3)_x$ with $x = 0$–0.6 (Supplementary Fig. 5). **f** An intermediate layer structure for the flat surface (case 1), the wrinkled structure (case 2) and the coarse surface with trace of wrinkle (case 3). A bilayer model with a solid upper layer with thickness of $h_f$ was proposed to explain the wrinkling process. **g** Schematic representation of setup for measuring optical diffraction pattern at bottom surface of bilayer film. **h** Reflected optical diffraction patterns for bilayer and monolayer films. **i** Photographs of reflected (up), transmitted (down) optical diffraction patterns as function of spin-coating time (11, 12, 14, and 16 s) after perovskite precursor is contacted with diethyl ether at 10 s ($T_{Sub} = 15\,°C$ and $T_{DE} = 5\,°C$).

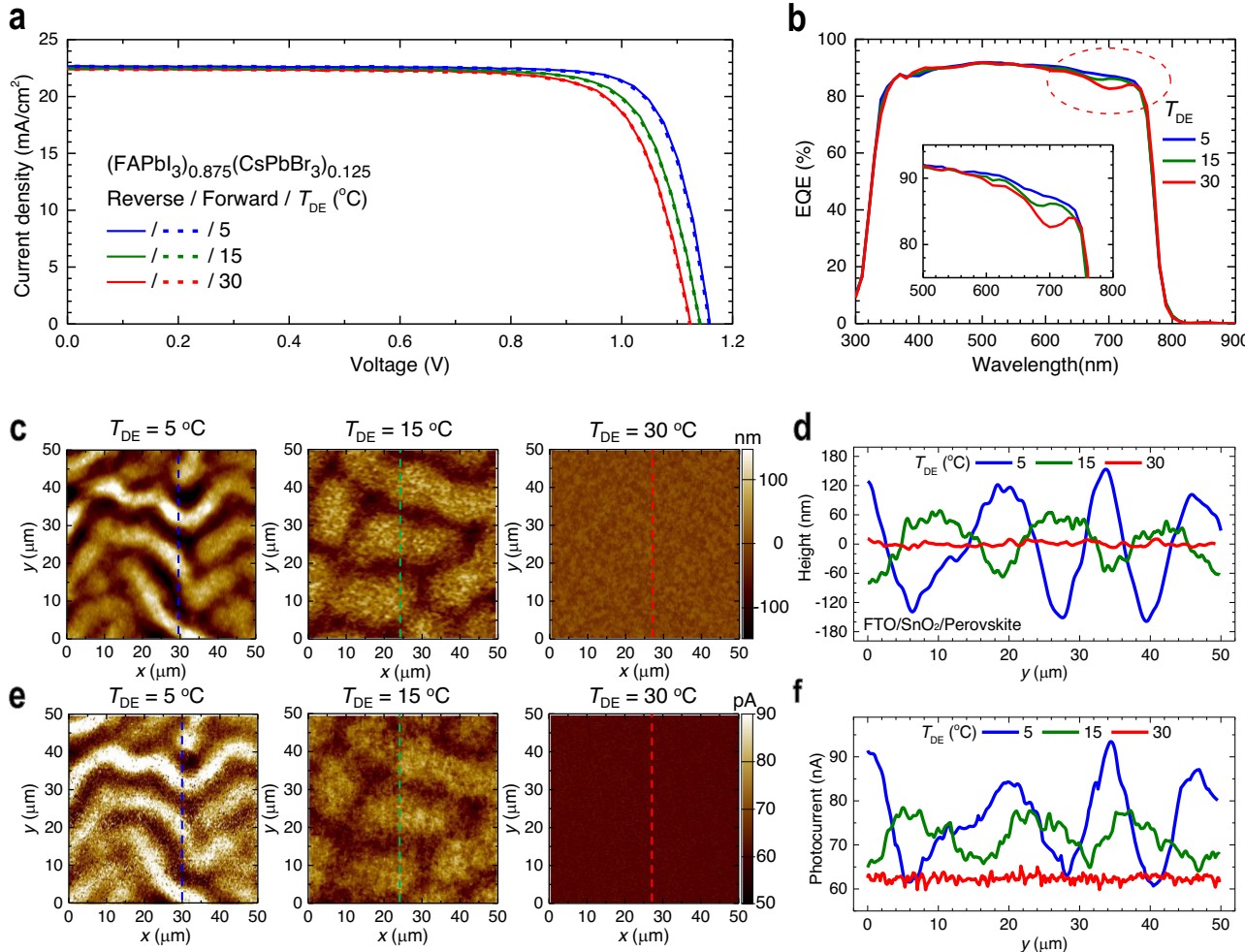

**Fig. 3 Dependence of current-voltage curves on $T_{DE}$ and photocurrent depending on the hills and the valleys. a** Current density ($J$) – voltage ($V$) curves of PSCs employing $(FAPbI_3)_{0.875}(CsPbBr_3)_{0.125}$ films formed from different $T_{DE}$. Solid and dashed lines represent reverse and forward scanned data, respectively. **b** External quantum efficiency (EQE) for each condition. Inset shows EQE from 500 to 800 nm. **c** Atomic force microscopy (AFM) topographies of the annealed $(FAPbI_3)_{0.875}(CsPbBr_3)_{0.125}$ films deposited on the SnO$_2$-coated FTO substrate at $T_{Sub} = 15$ °C and at different $T_{DE}$ of 5, 15, and 30 °C. $\lambda$ and $A$ are estimated to be 13.4 μm and 120 nm for $T_{DE} = 5$ °C, 17.1 μm, and 62 nm for $T_{DE} = 15$ °C. **d** Height profile with respect to the contour lines in **c**. **e** Photoconductive AFM (pc-AFM) images of the annealed $(FAPbI_3)_{0.875}(CsPbBr_3)_{0.125}$ films deposited on the SnO$_2$-coated FTO substrate at $T_{Sub} = 15$ °C and at different $T_{DE}$ of 5, 15, and 30 °C on the same areas in **c**. **f** pc-AFM data-based photocurrent profile with respect to the contour lines in **c** or **e**.

Table 1 and Supplementary Fig. 10, the forward- and reverse-scanned photovoltaic measurements exhibited improvement of the overall average PCE from 19.46% ($J_{sc}$: 22.365 mA/cm$^2$, $V_{oc}$: 1.123 V, FF: 0.7729) to 21.00%. ($J_{sc}$: 22.662 mA/cm$^2$, $V_{oc}$: 1.158 V, FF: 0.8004) upon decreasing $T_{DE}$ from 30 to 5 °C. In Fig. 3b, we can identify the possible contributions made by the wrinkled morphology. For example, in the spectral range between 600 and 750 nm, EQE shows enhancement for the wrinkled morphology ($T_{DE} = 5$ °C) compared to the flat reference ($T_{DE} = 30$ °C). This enhancement can be attributed by extended collection length ($L_C$) of carriers responsible for the longer wavelength incident photons. The extended charge carrier collection length corresponds to the longer carrier diffusion length ($L_D$)[23]. This implies that the enhanced photovoltaic performances of the wrinkled morphology is due mainly to the facilitated transport property of photocarriers. To verify the facilitated transport, microscopic pc-AFM images are measured for the wrinkled sample. As shown in Fig. 3c–f, pc-AFM images clearly exhibit the strong correlation between the wrinkle morphology and the photocurrent. In a device configuration with FTO/SnO$_2$/Perovskite/spiro-MeOTAD

layers, the spiro-MeOTAD layer also exhibits a wrinkle morphology (Supplementary Fig. 11), which indicates that spiro-MeOTAD layer forms conformal contact with the wrinkled perovskite thin film without delamination ($A$ is reduced to 70 nm for $T_{DE} = 5$ °C and 25 nm for $T_{DE} = 15$ °C). As shown in Fig. 3f, it is remarkable that the photocurrent is higher for the hill sites with lower $T_{DE}$, while photocurrent for the valley sites is nearly constant regardless of $T_{DE}$. The average photocurrent (over the area of $50 \times 50$ μm$^2$) increases from 62.1 to 69.6 pA and to 76.0 pA as $T_{DE}$ decreases from 30 to 15 °C and to 5 °C. To study the dependence of perovskite thickness on the measured photocurrent, $J$–$V$ curves and pc-AFM are measured for the flat $(FAPbI_3)_{0.875}(CsPbBr_3)_{0.125}$ films with different film thickness of 477 and 693 nm. As shown in Supplementary Fig. 12, the difference in the measured $J_{sc}$ and in the pc-AFM photocurrent is negligible, i.e., $\Delta J_{sc} = 0.5$ mA/cm$^2$ and $\Delta$photocurrent = ~2.9 pA, respectively. This indicates that the photocurrent difference at the hill and valley sites is not due to the thickness difference. In addition, it was reported that the concentration of photogenerated carriers is reduced to about 1/10 at a depth of

412 nm under the film surface and to 1/100 at 709 nm as compared to the carrier concentration at near surface (~67 nm)[24]. Therefore, it is expected that difference in photocurrent between 477 nm- and 693 nm-thick perovskite layers is <4%. These findings allow us to explore another mechanism for the photocurrent enhancement given by the micro wrinkle morphology (i.e., ~470 nm (Valley, $T_{DE} = 5\,°C$) and ~700 nm (Hill, $T_{DE} = 5\,°C$)).

**Carrier lifetime of wrinkled perovskite film.** One of the possible contributions of the wrinkled morphology to the photocurrent enhancement can be found in better crystallinity of the film. Indeed, although the wrinkling process seems to be hardly changes the grain size, an enhanced intensity of the ($h$00) planes with decreasing $T_{DE}$, as could be found from X-ray diffraction measurements (Supplementary Fig. 13), is indicative of a more preferred orientation of the grains, which could be advantageous to charge extraction[25]. To find the underlying mechanism of the enhanced photocurrent and overall photovoltaic performances of the wrinkled morphology perovskite films, we have attempted to analyze microscopic dynamics of the photocarriers inside the films at the hill and valley sites. Using fluorescence lifetime imaging microscopy (FLIM) coupled with time-correlated single-photon counting (TCSPC), we measure position-specific lifetime map of the photocarriers in the wrinkled perovskite films. For better comparison, FLIM images are measured from top and from the bottom side of each perovskite film. As shown in Fig. 4a, b, it is evident that the wrinkle morphology ($T_{DE}$ of 5 and 15 °C) is distinctively and consistently reflected in the microscopic PL lifetimes maps, where valleys appear brighter than hills (hill and valley sites in FLIM are identified from the optical microscope images of the same areas (Supplementary Fig. 14). No compositional changes between hill and valley on top and bottom sides are confirmed by steady-state photoluminescence (PL) (Supplementary Fig. 15), which is consistent with previous report[26]. We associate this to the differences in the local layer thickness of hill and valley, so that the laser induced charge carriers distribute within a larger volume in case of the hills, leading to a lower charge carrier density and thus lower PL intensity. In particular, when comparing the average lifetime of the photocarriers between top and bottom, it is also evident that the higher the amplitude of the wrinkles, the greater the difference between the lifetimes of the carriers at the hill and valley sites. This can also be confirmed by the nearly constant lifetime maps of the flat sample (i.e., film with $T_{DE} = 30\,°C$ in Fig. 4a, b). In detail, we analyze the average lifetime of the photocarriers at the hill and valley sites with time-resolved PL (TRPL) as shown in Fig. 4c, d. In general, the measured PL decay at earlier times after excitation is dominated by bimolecular recombination, while for longer times, it is dominated by monomolecular recombination at defects[27]. The latter process is strongly correlated with the defect concentration[28]. From the exponential fitting of the PL decay curve, we can deduce the decaying rate $k_1$ such that $PL(t) \propto \exp(-2 \cdot k_1 t)$ (see Supplementary note 4 for details and Supplementary Table 2). From the exponential fitting, we found that $k_1$ is lower at the hill sites, while higher at the valley sites. Also, it decreases as $T_{DE}$ is lowered (i.e., $3.6 \times 10^6\,s^{-1}$ or $4.4 \times 10^6\,s^{-1}$ for the spots T-a or T-c (hill sites) vs. $5.7 \times 10^6\,s^{-1}$ or $7.4 \times 10^6\,s^{-1}$ for the spots T-b or T-d (valley sites)). The difference in the recombination rate at the hill and valley sites can be attributed by the difference of the local defect densities at the hill and the valley sites. It was reported that both tensile, as well as compressive strain in halide perovskite thin films lead to an increase in the defect density[29]. Areas with higher local strain can result in faster PL decay[30]. Atomistic calculations based on the first-principle models, the defect density is indeed proportional to the degree of the local strain. Indeed, $k_1$ is observed to

decrease with higher amplitude wrinkle morphology (with lower $T_{DE}$), which indicates that defects densities at the hill sites decrease with amplitude. This can be attributed to the reduced structural defects such as grain boundary defects at the hill sites because local strain is additionally alleviated at structural defects[31]. The reduced grain boundary defects should be accompanied by the enhanced uniformity of the grain sizes, which can be checked by the narrower distribution of the grain areas (Supplementary Fig. 16). Therefore, it is possible to suggest that the wrinkle morphology provides additional virtue for the hill sites with less defects, which in turn is responsible for the extended lifetime of the photocarriers, which results in the higher photocurrent at pc-AFM measurement. This can be confirmed again by the comparison of the photocurrent and average carrier lifetime as shown in Fig. 4e. As compared in the plot, the average diffusion length of the photocarriers, which is proportional to the square root of the lifetime of top and bottom surface, $\tau_{av}^{1/2}$, is strongly correlated with photocurrent. Considering a fact that the lifetime is inversely proportional to defect density ($N_d$), we can deduce that the hill sites with higher amplitude is clearly associated with lower defects than flat or valley sites. Moreover, it is also notable that $k_1$ is lower at the bottom than at the top (i.e., $k_1 = 3.6 \times 10^6\,s^{-1}$ in spot T-a at the top vs. $k_1 = 2.6 \times 10^6\,s^{-1}$ in spot B-f at the bottom). This indicates the bottom side of the films has the lower defect concentration. This would confirm again that that the crystal grows from the top surface (initially crystallized part with more defects) to the bottom (retarded crystallization in relatively DMSO-rich environment), which can further allow lower defects. Based on these observations, we can suggest that the microscopic wrinkle morphology genuinely gives rise to reduced defects, which considerably extends photocarrier lifetimes, and therefore, increase the $V_{oc}$ and FF. The decreased defect densities facilitate charge extraction at the perovskite/electron transport layer (ETL) interface and results in a gain in voltage.

**Light soaking stability.** Based on the analysis of wrinkling mechanism and the wrinkle-morphology effects on the optoelectronic properties, we can proceed to the morphology tailoring to maximize the photovoltaic performances. We have found the optimized composition for the best wrinkle morphology such as $FA_{0.92}Cs_{0.08}PbBr_{0.15}I_{2.85}$ including a K-doping approach to passivate halide ion interstitials due to Frenkel defects[32,33], where the K-doped $FA_{0.92}Cs_{0.08}PbBr_{0.15}I_{2.85}$ results in $\lambda = 14\,\mu m$ and $A = 115\,nm$ at $T_{DE} = 5\,°C$. With this tailored perovskite thin films, we obtain a PCE as high as 23.02% along with $J_{sc} = 23.536\,mA/cm^2$, $V_{oc} = 1.1948\,V$ and FF = 0.8188 (Supplementary Fig. 17). This optimized morphology also exhibit satisfactory performances of long-term stability. As shown in Supplementary Fig. 18, from the long-term light soaking test for over 1000 h conducted from maximum power point tracking (MPPT) under continuous light illumination (97 mW/cm²), stable performance as high as 83.4% of the initial PCE (22.09% (0 h) to 18.42% (1008 h)) is maintained. The decrease in PCE by 11.6% until 432 h (from 22.09 to 19.74% (432 h)) is due mainly to a lowered $V_{oc}$ without alternation in $J_{sc}$ (see inset in Supplementary Fig. 18), which might be attributed to interfacial defect generated during illumination[34,35], after which the photovoltaic parameters remain almost unchanged as confirmed by $J$–$V$ curves.

In this study, we found the substantial effects of the microscopic wrinkles on the photovoltaic performances of the perovskite solar cells. The wrinkling mechanism hinges on the compressive stress relaxation of the bilayer in the course of spin-coating intervened by the antisolvent-driven microscopic phase separation of the film. The wrinkle geometry such as wavelength and amplitude were systematically controlled by changing the composition, antisolvent

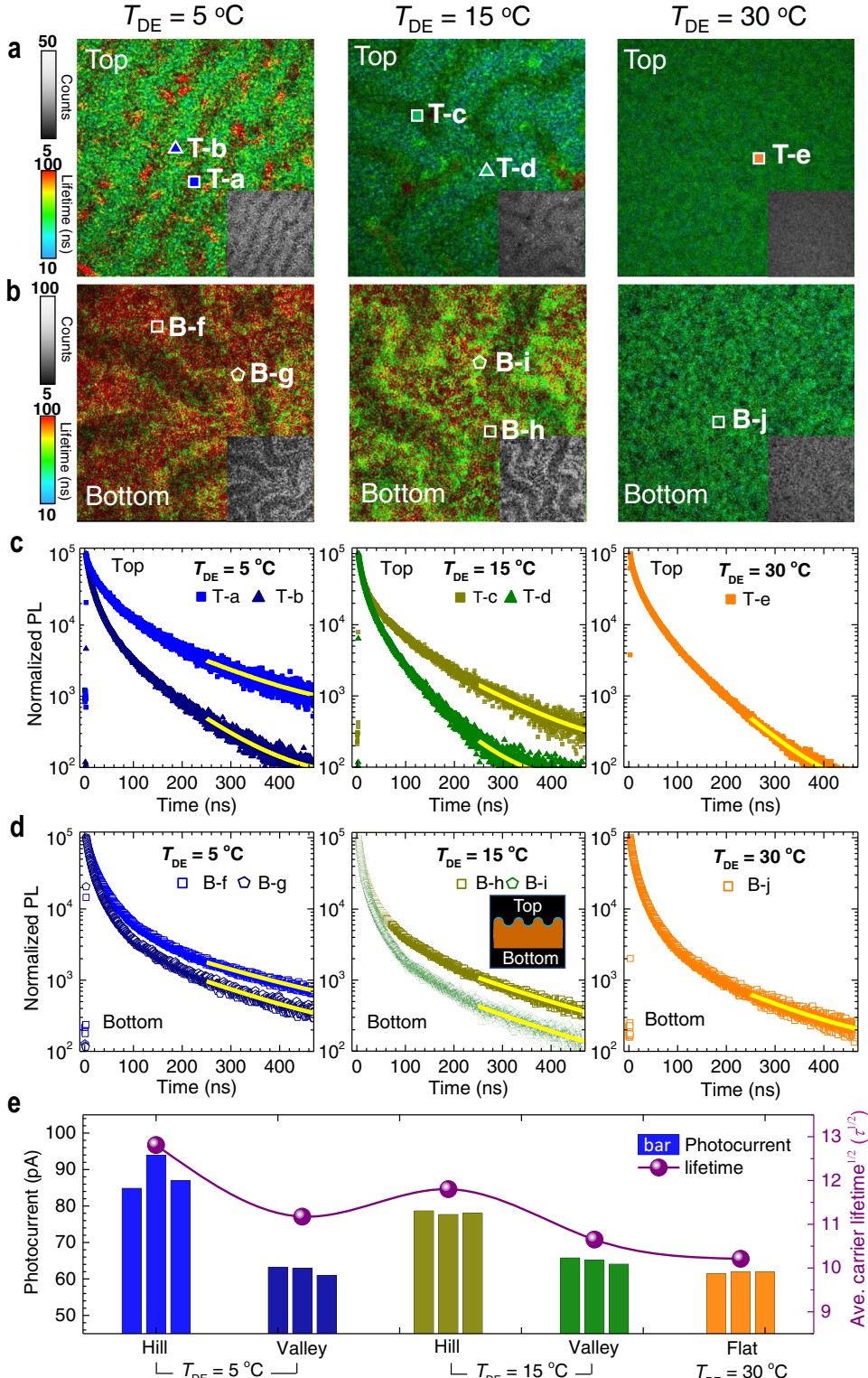

**Fig. 4 Carrier lifetime on the hills and the valleys on top and bottom of the sinusoidal wrinkled structures.** Fluorescence lifetime imaging microscopy (FLIM) images on (**a**) the top and (**b**) the bottom of the PMMA/(FAPbI$_3$)$_{0.875}$(CsPbBr$_3$)$_{0.125}$/Glass device, where the perovskite layers were prepared at $T_{DE}$ = 5, 15, and 30 °C at the given $T_{Sub}$ = 15 °C. PMMA (poly(methyl methacrylate)) and glass are top and bottom side, respectively. Insets are gray scale images. Plots of time-correlated single-photon counting (TCSPC) measured on the selected areas (hill (T-a, B-f, T-c, and B-h), valley (T-b, B-g, T-d, and B-i) and flat (T-e and B-j) on (**c**) the top and (**d**) the bottom surface, where top and bottom sites are illustrated in inset. Imaging area was 50 × 50 μm² and each pixel was about ~97 nm in diameter (excitation/PL collection spot: 512 × 512 pixels). **e** Plot of $\tau_{av}^{1/2}$ ($\tau_{av} = (\tau_{hill} + \tau_{valley})/2$) and photocurrent values from pc-AFM at hill and valley (three positions of each area) in Fig. 3e of the PMMA/(FAPbI$_3$)$_{0.875}$(CsPbBr$_3$)$_{0.125}$/Glass device.

temperature and substrate temperature. We provided a fundamental analysis based on the theoretical model supported by experiments to elucidate the wrinkling mechanism. In addition, we also provided the detailed mechanism underlying the microscopic wrinkle-driven enhancement of photocurrents. From the experimental observations, we found that the hill sites with higher amplitude of the wrinkles suffer less defects, which in turn extended photocarrier lifetime, which results in enhanced photo-response and $V_{oc}$. The wrinkling process was also found to be beneficial to both ETL and hole transport layer (HTL) interfaces because of a decreased defect concentration in the perovskite. Based on the understanding the fundamental mechanism of the wrinkling and its effects on the photovoltaic performances, we tailored the morphology to maximize the photovoltaic properties, and could obtain a PCE as high as 23% with satisfactory long-term stability. We expect the present study delivers a substantial benefit in further exploring the possibility of enhancement of photovoltaic performance and stability by tailoring photocarriers via wrinkling the perovskite films.

## Methods

**Materials synthesis.** Formamidinium iodide (FAI, FA = HC(NH$_2$)$_2$$^+$) or methylammonium iodide (MAI, MA = CH$_3$NH$_3$$^+$) was synthesized by reacting 20 mL hydroiodic acid (57 wt% in water, Sigma Aldrich) with 10 g of formamidinium acetate (99%, Sigma Aldrich) or 18.2 mL of methylamine (40 wt% in methanol, TCI) in an ice bath. After stirring for 30 min, brown precipitate was formed, which was collected by evaporating the solvent at 60 °C using a rotary evaporator. The solid precipitate was washed with diethyl ether (99.0%, Samchun) several times, followed by recrystallization in anhydrous ethanol. The white precipitate was dried under vacuum for 24 h and then stored in a glove box filled with Ar.

**Device fabrication.** The patterned FTO glass (Pilkington, TEC-8, 8 Ω/sq) was ultrasonically cleaned with detergent, DI water, ethanol and acetone, successively, which was treated with Ultraviolet–Ozone (UVO) for 40 min to remove organic contaminants. The 15 wt% SnO$_2$ aqueous colloidal solution (Alfa Aesar) was diluted to 4 wt%, which was spin-coated on the FTO-coated glass substrates at 3000 rpm for 30 s and then annealed on a hot plate in ambient air atmosphere at 185 °C for 30 min. After cooling down to room temperature, the film was exposed again to UVO for 40 min prior to coating the perovskite layer. Perovskite films with different compositions were spin-coated using precursor solutions. For example, the (FAPbI$_3$)$_{0.875}$(CsPbBr$_3$)$_{0.125}$ perovskite was deposited using a precursor solution prepared by mixing 0.1505 g of FAI, 0.4034 g of PbI$_2$ (99.9985%, Alfa Aesar), 0.0459 g of PbBr$_2$ (99.999%, Sigma Aldrich) and 0.0266 g of CsBr (99.999%, Sigma Aldrich) in 75 μL dimethyl sulfoxide (DMSO, > 99.5%, Sigma Aldrich), 0.525 mL of N,N'-dimethylformamide (99.8% anhydrous, Sigma Aldrich). The solutions were filtered with 0.20 μm-pore-sized PTFE-H filter (Hyundai MICRO). Prior to coating, the precursor solutions were mildly stirred at 50 °C for 10 min in ambient condition to remove unwanted gas molecules dissolved in the solutions. The precursor solutions (solution temperature was 15 °C) were spin-coated on the substrate (substrate temperature was 5 °C or 15 °C) at 4000 rpm for 20 s, where 0.35 mL of diethyl ether with different temperature (5, 15, or 30 °C) was dripped in 10 s after spinning. The brownish adduct films were formed right after deposition, which was heated at 145 °C for 10 min. The 20 μL of spiro-MeOTAD solution, which was prepared by dissolving 72.3 mg spiro-MeOTAD, 28.8 μL of 4-tert-butyl pyridine and 17.5 μL of lithium bis(trifluoromethanesulfonyl)imide (Li-TFSI) solution (520 mg Li-TSFI in 1 mL acetonitrile (99.8%, Sigma Aldrich)) in 1 mL of chlorobenzene, was spin-coated on the perovskite layer at 3000 rpm for 20 s. Whole coating process was carried out in dry room (RH < 0.3%, 18–20 °C). Finally, 90 nm of Au electrode was deposited by using a thermal evaporator at an evaporation rate of 0.05 nm/s. For K-doped FA$_{0.92}$Cs$_{0.08}$PbBr$_{0.15}$I$_{2.85}$ perovskite, a precursor solution was prepared by mixing 0.1592 g of FAI, 0.4449 g of PbI$_2$ (99.9985%, Alfa Aesar), 0.0128 g of PbBr$_2$ (99.999%, Sigma Aldrich) and 0.0170 g of CsBr (99.999%, Sigma Aldrich) in 75 μL dimethyl sulfoxide (DMSO, > 99.5%, Sigma Aldrich), 0.465 mL of N,N'-dimethylformamide (99.8% anhydrous, Sigma Aldrich) and 60 μL KI solution (0.166 g of KI (99.999%, Sigma Aldrich) in 10 mL of DMF). The precursor solution kept at temperature of 15 °C was spin-coated on the substrate (substrate temperature was 15 °C) at 4000 rpm for 20 s, where 0.35 mL of diethyl ether ($T_{DE}$ = 5 °C) was dropped in 10 s after spinning. The brownish adduct film formed right after deposition was heated at 145 °C for 10 min. After the perovskite film was cooled down to room temperature, 30 μL of 4-fluoro phenylethylammonium iodide (4F-PEAI) solution (10 mM in IPA) was spin-coated at 6,000 rpm for 20 s.

**Characterizations.** Optical images were obtained by using an inverted optical microscope (Primo Vert, Carl Zeiss) with objective lens (Primo Plan-ACHROMAT 20x/0.30). Current density-voltage (J–V) curves were measured under AM 1.5 G one

sun (100 mW/cm$^2$) illumination using a solar simulator (Oriel Sol 3A, class AAA) equipped with 450 W Xenon lamp (Newport 6280NS) and a Kiethley 2400 source meter. The light intensity was adjusted by NREL-calibrated Si solar cell having KG-5 filter. The device was covered with a metal mask with aperture area of 0.125 cm$^2$. The External Quantum Efficiency (EQE) spectra were collected by using an QEX-7 series system (PV measurements Inc.) in which a monochromatic beam was generated from a 75 W Xenon source lamp (USHIO, Japan) under DC mode. Steady-state photoluminescence (PL) were measured by a Quantaurus-Tau compact fluorescence lifetime spectrometer (Quantaurus-Tau C11367-12, Hamamatsu). The film samples were excited with 464 nm laser (PLP-10, model M12488-33, peak power of 231 mW and pulse duration of 53 ps, Hamamatsu) pulsed at repetition frequency of 10 MHz for steady-state PL. All measurements were done at room temperature (~298 K). Viscosity was measured by using rheometer (25 mm aluminum parallel plate, TA Instruments, New Castle, DE, USA) at different temperature with 0.6 rad/s for solid (Zero shear-rate viscosity) and 120 rad/s for liquid of angular frequency and oscillation strain 1%. The atomic force microscopy (AFM) and photoconductive AFM (pc-AFM) were measured with a perovskite/SnO$_2$/FTO and a spiro-MeOTAD/perovskite/SnO$_2$/FTO structured samples, respectively, by using white LED (light intensity: 0.228 mW/cm$^2$) and 1 V of sample bias (NX10 system, AD-2.8-AS ($k$ = 2.8 N/m, conductive diamond coating, radius = 10 nm) or CDT-CONTR ($k$ = 0.5 N/m, conductive diamond coating, Park Systems). All images were obtained under the ambient condition. Ellipsometry measurements were carried out with the perovskite/c-Si, the SnO$_2$/c-Si and the spiro-MeOTAD/c-Si samples by using Elli-SE Ellipsometer from 240 to 1000 nm (1.2–5.2 eV) wavelength. In situ photoluminescence (PL) and UV–Vis absorption were measured while spinning the samples by custom-built setup. In detail, brushless DC motor (Trinamic BLDC4208) was connected to the rotatable chuck that was connected with Peltier element located on a heat diffusor via a cogged V-belt. RPM was controlled with a motor driver (Trinamic TMCM-1640). To allow quasi-simultaneous detection of both absorption and PL during processing, a diode laser (520 nm) was supplied on the substrate and white light (generic cold white LED) was incident above the substrate. The optical fiber was used for gathering the signals. By using a mechanical chopper with blade and mirror, light path was separated and gathered spectrometers for absorption and for PL. The fluorescence lifetime imaging microscopy (FLIM) measurement was carried out on a PicoQuant MicroTime 200. The FLIM system is based on an inverted optical microscope (Olympus IX71). A pulsed laser source with 561 nm (PicoQuant) was operated at 2.5 MHz and a fluence of about 0.5 μJ/cm$^2$. The laser was focused on the sample through a high numerical aperture objective lens (Olympus PlanApo 60×/1.20 water immersion and 100×/0.9 air). The emission from the sample was passed through a long-pass filter and a 100 μm pinhole before being detected by a single-photon avalanche diode (SPCM-AQRH SPAD, Excelitas Technologies) and processed by time-correlated single-photon counting (TCSPC) electronics (Time harp 260 Pico, PicoQuant). The FLIM images were analyzed by PicoQuant SynPhoTime 64 (v. 2.4.4874).

**Long-term stability test.** Long-term light soaking test for over 1000 h was conducted by exposing the unsealed device to a white LED light with intensity of 97 mW/cm$^2$ (0.97 sun) in N$_2$ glove box at temperature ranging between 25 and 31 °C, where UV filter (Schott, GG-400) was applied to the device and pre-conditioning was performed before light soaking experiment by aging the fresh device for 96 h under 0.6 mW/cm$^2$. The J–V curves and the steady-state PCE at maximum power point tracking (MPPT) were measured every 12 h or 24 h in dry room with relative humidity of <5% using a solar simulator (VeraSol-2 LED Class AAA Solar Simulator (Newport), 100 mW/cm$^2$). After each measurement, the devices were stored under 0.97 sun illumination in N$_2$ globe box again. The metal mask with aperture area of 0.10 cm$^2$ was placed on top of the cell.

**Reporting summary.** Further information on research design is available in the Nature Research Reporting Summary linked to this article.

## Data availability
The authors declare that the main data supporting the findings of this study are available within the article and its Supplementary Information files. Extra data are available from the corresponding author upon reasonable request.

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

## Acknowledgements
This work was supported by the National Research Foundation of Korea (NRF) grants funded by the Ministry of Science, ICT Future Planning (MSIP) of Korea under contracts NRF-2012M3A6A7054861 (Global Frontier R&D Program on Center for Multiscale Energy System) and NRF-2016M3D1A1027663 and NRF-2016M3D1A1027664 (Future Materials Discovery Program). This work was supported in part by Basic Science Research Program through the NRF under contract NRF-2018K2A9A2A15075785 (Germany-Korea Partnership Program). S.-G.K. acknowledges financial support from NRF under contact 2016R1A2B3008845, NRF-2017H1A2A1046990 (NRF-2017-Fostering Core Leaders of the Future Basic Science Program/Global Ph.D. Fellowship Program). Corresponding support was provided by the German Academic Exchange Service (DAAD project-ID 57449733). S.H. and P.R. thank the German Science Foundation (DFG) for financial support and the BPI KeyLab Electron and Optical Microscopy. S.H. thanks the Bavarian framework program Soltech for funding. K.S. acknowledges financial support from the German National Science Foundation (Project KO 3973/2-1 and GRK 1640). F.P. acknowledges support by the German National Science Foundation via the Project PA 3373/3-1. Y.Z. acknowledges funding from China Scholarship Council. The authors thank PicoQuant Inc for providing SynPhoTime 64(v. 2.4.4874) Program.

## Author contributions
N.-G.P., F.P., and S.H. supervised the research; S.-G.K. designed and conducted experiments and measurements; J.-H.K. conducted experiments and measurements; Y.Z., S.-G.K., and K.S. measured and analyzed in situ PL and absorbance measurements. P.R. and S.-G.K. measured and analyzed FLIM and TCSPC. S.J.K. conducted simulation study of bilayer wrinkle system. S.-G.K. and N.-G.P. wrote manuscript; and all authors discussed the results and revised the manuscript.

## Competing interests
The authors declare no competing interests.
