## [Peer Review File · Nature Communications]

REVIEWER COMMENTS

Reviewer #1 (Remarks to the Author):

Many research groups have seen variations in the thickness of their perovskite films and a couple of groups have written specifically on the subject. The topic is important because smooth films make it easier to deposit contact layers that have uniform thickness. Although most people seem to want smooth films, the authors of this manuscript obtained higher efficiency in solar cells with the wrinkling.

I really like the new data in this manuscript. They have made films with many different compositions using several different processing conditions. They observed interesting trends that I doubt many people could have predicted. I definitely think this data should be published, but at this time I find the explanations for the observations not to be convincing. If the authors address my comments, then the manuscript might be important enough for Nature Communications.

I find the statement "no systematic studies were carried out to understand the formation mechanism of a pseudo-epitaxial wrinkle morphology depending on composition and/or preparation condition" to be misleading. Reference 20 did far more than simply report that wrinkling exists. It provided x-ray diffraction and wafer curvature stress measurements at several stages in the film formation process to show how compressive stress causes the wrinkling. It also showed how changes in the solvent composition could mitigate the wrinkling. I think that the authors are building on the explanation in Reference 20 by adding that the wrinkling can more easily occur in a perovskite layer that rests on top of a viscous layer.

I think the claims that the wrinkling improves light management are very misleading. It is well known that texturing can help trap light in solar cells. In this case, however, there are approximately 60 nm variations in height with periodicities of around 15 microns. The schematics are not to scale and greatly exaggerate the extent of the surface height variation. I wouldn't expect such a small change in the surface angle to help much. The EQE plots and the change in J_{sc} support my position. At best, the light trapping helps just a tiny bit around 700 nm. Reducing the temperature of the diethyl ether deposition clearly had its impact by increasing Voc, which is unlikely to be a result of light trapping.

On page 16, the authors describe the FDTD simulations of how light will propagate in the solar cells. Their analysis of the data is simply "Light absorption is more enhanced in the wrinkled structure than in the flat one." If they properly analyze their FDTD data in a quantitative way and find out the percentage change in light absorption, I think it will support what I have said in the previous paragraph.

At the end of a paper that very thoroughly examines the effect of varying the temperature of the anti-solvent and the primary solvent composition on wrinkling and optoelectronic properties, the authors suddenly say that they raised the efficiency considerably by adding some KI, which has nothing to do with the rest of the paper. They don't actually say how much KI they added. I find that very unsatisfying. The authors must say how much additive they used. Instead of telling readers in the abstract that they reached 23 % efficiency (with KI), they should say what efficiency they obtained as a function of DE deposition temperature and explain that they get higher efficiency in the wrinkled films. That is the real take home message for this manuscript.

I do not understand why the authors say "Since the wrinkle texture certainly has the potential for epitaxial growth due to an underlying buckling process associated with a compressive stress." I don't see what epitaxial growth has to do with buckling or compressive stress. The authors say this work is motivated by obtaining epitaxial growth. I only see one line on page 16 that says that an XRD plot in the supplemental section shows better orientation for the low temperature antisolvent film.

The statement on page 14 that "This indicates lower defect concentrations at the bottom side of the films, which is attributed to the crystal growth direction from the top surface to the bottom." is not clear or well supported by data. The authors reference Stranks et al's manuscript (ref 30) on

how stress affects trap density and photoluminescence, but do not say anything about it. I recommend summarizing the main points of that manuscript and using them to try to explain the observations reported in this manuscript. Why do the authors think stress affects the trap density?

In Fig 4, the authors have both a grey scale and a color scale. I don't see how one can have two color scales for a 2D plot. We cannot see the grey scale.

I don't think "Topographical tailoring" are good words to use in the title. I was partially expecting the paper to be about topological insulators, which it isn't at all. People won't know what the authors mean by "topographical tailoring." I suggest a title such as "How perovskite composition and antisolvent miscibility affect film wrinkling and optoelectronic properties."

I think that the choice of solvents is so important that it should be stated clearly early in the main text, not just in the supplemental section. The way the manuscript is written, one might think that DMSO is the primary solvent. In fact, it is an additive and DMF is the main solvent. The authors might also want to clarify that the DMF probably evaporates first, which is probably why they write about the DMSO more.

On page 4, the authors refer to "blue frames" in Fig 1. It took me a while to notice the blue frames, which are very hard to see. It is slightly hard to follow the trends in how composition affects wrinkling by looking at microscope images, many of which are too small. I really like the plots of λ vs composition in Fig S1 d and e. I recommend putting those plots in the main text. They helped me understand the trend instantly.

On page 10 (line 218), should "to form lower TDE" be "to form at lower TDE?"

Why were the substrates at 15 degrees C during spin casting? That is unusual and inconvenient. There must be a good reason.

Reviewer #2 (Remarks to the Author):

Park et al. reported a universal method to produce a pseudo-epitaxial perovskite layer via manipulating strain and compression relaxation in a perovskite film, which lead to a controllable wrinkled film structure. Deep study revealed the wrinkling mechanism of the compressing tensile-strained surface through a bilayer intermediate. Finally, the finding of hill sites showed longer carrier lifetimes and higher optical absorption than the valley sites, providing the basis for over 23% PCE and long-term operational stability of PSCs. This is an interesting and deep work on growth kinetics control of perovskite crystals, and the connection between topological arrangement and photocarriers behavior provides useful guidance for more efficient and stable PSCs design. However, there are still some concerns, which need more clarification:

(1) The wrinkling structure only occurs under some specific components (i.e. (FAPbI₃)_{0.875}(CsPbBr₃)_{0.125}) at given conditions (i.e. TDE is as low as 5 °C), thus it is hard to popularize this method and the cooling of diethyl ether will increase the production costs. Actually, there have been some components which may hardly form wrinkling structure (Advanced Materials, 2020, 32, 1907757; Science, 2019, 366, 749–753; Nature Photonics, 2019, 13, 460–466, etc.) but exhibit higher PCE and similar stability compared to this work, and all of them are fabricated under simpler process. The authors should comment on this fact and the potential advantages of this method.

(2) The bilayer intermediate is the key for wrinkling structure formation, however, whether such an intermediate will bring into the component inhomogeneity along with the vertical direction of the perovskite films? Moreover, the existence of wrinkling structure may also lead to the component inhomogeneity between hills and valleys. The authors should provide more information on these details.

(3) The amplitude (A) is about 100 nm for optimized perovskite film, although this value may have no obvious influence on Spiro-OMeTAD deposition (which obtains a thickness over 200 nm),

however, some more stable HTLs such as PTAA, P3HT is hard to form continuous film when depositing on the wrinkle perovskite, resulting to the existence of lots of shunting paths. This is another concern that hinders the development of this method.

(4) In-situ absorption data showed a perovskite phase with thickness of 250 nm for TDE = 5 °C, 362 nm for TDE = 15 °C and 450 nm for TDE = 30 °C. Since anti-solvent was dropped on the top surface, it was reasonable to think that the pre-formed perovskite phase had a trend to distribute near the surface. But this evidence is not conclusive for a bilayer structure because partial pre-formed perovskite phase might also locate in bulk which could not be fully excluded so far. So, are there further characterizations or explanations on this especially a direct evidence of phase boundary?

(5) How to understand the decreased defect densities in the wrinkle structure compared with the flat morphology? Moreover, why the hill sites show longer carrier lifetimes than valley sites? The authors should give more description on the deep mechanism.

(6) A minor question is that some calculation details can be put into SI, and the description of the connection between perovskite components and wrinkling structures can be more concise so that the readers can easily get the important information.

Reviewer #3 (Remarks to the Author):

This work by S.-G. Kim et al reports a systematic study on the wrinkle-like texture formation in a series of halide perovskite thin films. The structure and morphology of the wavy texture was controlled by varying the anti-solvent and substrate temperatures. The formation mechanism was investigated using a series of characterization tools such as in-situ PL and XRD. Using these textured perovskite thin film, solar cells with high efficiency up to 23% was fabricated. Although the formation of the textured surface has been reported by a number of groups previously, this work is more comprehensive and contains some interesting results. However, I feel that there are large amounts of inconsistency and many of the claims in this paper can not be supported by the experimental data. My specific points are listed below. Because of this, I do not recommend publication of this work, at least in the present form.

Specific points for the authors:

1. The authors mentioned epitaxial growth several times, however, there is no epitaxial growth in this work.
2. Regarding the growth mechanism: a. I can not understand where the lattice strain comes from as the growth from the TiO₂ layer is not epitaxial. b. Then it is also hard to understand why the "wrinkle" surface can release the strain as the wrinkle is very much macroscopic with a wavelength on the order of micrometers. c. Furthermore, Fig 3a and 3b seems no difference to me. The peaks at low angles are so broad. It is unclear to me how the peak values were selected and how Fig 3c was generated. d. The in-situ PL does not provide constructive information regarding the formation mechanism. e. There is no experimental evidence for the bi-layer model.
3. Regarding Fig 4: It can be seen that in general the lower diethyl ether dripping temperature lead to longer lifetime for the whole film, and the lifetime for the bottom side of the film is better than top, but I do not see the "hill" is better than the "valley".
4. Line 320: the authors claim that the bottom of the film has lower defect density because the crystals growth from the top to bottom. I cannot understand why growth from top to bottom will lead to lower defect density at the bottom. More explanation and clarification are necessary.
5. Line 349: regarding the conducting AFM, the higher photoresponse may be from the larger thickness (more light absorption) at the "hill" sites over the "valley" sites. To me, this data is not convincing enough to prove the hill is better than the valley.

6. Based on the optical simulation, the light absorption is greatly enhanced in the textured film, but there is almost no difference in the device's photocurrent density.

7. Finally, the authors did not provide experimental details of the $\text{FA}_{0.92}\text{Cs}_{0.08}\text{PbBr}_{0.15}\text{I}_{2.85}$ solar cell fabrication in the method part. It is also confusing why the authors study the $\text{FA}_{0.875}\text{Cs}_{0.125}$ film in Figure 4 and part of Figure 5 and then switch to $\text{FA}_{0.92}\text{Cs}_{0.08}$ for solar cell fabrication.

Response to Reviewers Letters

Manuscript ID: NCOMMS-20-36179 -T

Title: Topographical tailoring of photocarriers in perovskite solar cell

Authors: Seul-Gi Kim¹, Jeong-Hyeon Kim¹, Philipp Ramming^{2,3}, Yu Zhong^{2,3}, Konstantin Schötz³, **Seok Joon Kwon**⁴ Sven Huettner², Fabian Panzer³, Nam-Gyu Park^{1*}

First of all, we thank the reviewers for their valuable comments on our manuscript of MS ID: NCOMMS-20-36179 -T entitled “Topographical tailoring of photocarriers in perovskite solar cell” (corresponding author: Nam-Gyu Park). Here we have addressed the queries from the reviewers and revised the manuscript according to the reviewers’ comments. The revised parts were highlighted in green in the revised manuscript. In addition, Dr. Seok Joon Kwon was included as co-author because he did numerical simulation of wrinkled morphology.

REVIEWER COMMENTS

Reviewer #1

Many research groups have seen variations in the thickness of their perovskite films and a couple of groups have written specifically on the subject. The topic is important because smooth films make it easier to deposit contact layers that have uniform thickness. Although most people seem to want smooth films, the authors of this manuscript obtained higher efficiency in solar cells with the wrinkling.

I really like the new data in this manuscript. They have made films with many different compositions using several different processing conditions. They observed interesting trends that I doubt many people could have predicted. I definitely think this data should be published, but at this time I find the explanations for the observations not to be convincing. If the authors address my comments, then the manuscript might be important enough for Nature Communications.

(Answer) Many thanks for the encouraging comment on our work.

1. I find the statement “no systematic studies were carried out to understand the formation mechanism of a pseudo-epitaxial wrinkle morphology depending on composition and/or

preparation condition“ to be misleading. Reference 20 did far more than simply report that wrinkling exists. It provided xray diffraction and wafer curvature stress measurements at several stages in the film formation process to show how compressive stress causes the wrinkling. It also showed how changes in the solvent composition could mitigate the wrinkling. I think that the authors are building on the explanation in Reference 20 by adding that the wrinkling can more easily occur in a perovskite layer that rests on top of a viscous layer.

(Answer) We thank the reviewer for the comment. We agreed with the reviewer's opinion and revised manuscript (MS) to reflect that opinion. MS is revised as “Recently, an approach to control the perovskite morphology has been explored. For example, microscopic wrinkles have been observed for a certain composition of perovskite that suffers buckling of the perovskite thin film [16,17]. In particular, the buckling was explained as a result of local compressive stress relaxation [17]. However, detailed and comprehensive studies for effects of the microscopic wrinkles on the photovoltaic performances as well as wrinkling mechanism have not been reported yet. Here, we report a simple and yet effective experimental approach to control and optimize the microscopic geometry of the wrinkles of perovskite thin films to maximize the photovoltaic performances as well as long-time durability. We also suggest a theoretical model elucidating the wrinkling mechanism based on the detailed experimental data.” in introduction part (p.3). And also, “Previous study on the wrinkling of perovskite thin films suggested that the compressive stress developed by the volume change during a fast perovskite formation led to wrinkling by using wafer curvature stress measurements [17]. This mechanism requires relatively long-time wrinkle formation dynamics up to several minutes to hours [19, 20]. However, we observe that the wrinkles form within 10 s. Therefore, we have developed a more detailed model by which the overall morphology of wrinkles can be elucidated as well as the wrinkling mechanism based on previous reports. [17, 18]. Using a model based on the thin film mechanics, the wrinkle geometry can be described as a function of the thickness and mechanical constants of the materials. We also derive relationships of λ and A of the wrinkles with the compositions and T_{DE} (see detailed analysis in SI).” in p.7.

2. I think the claims that the wrinkling improves light management are very misleading. It is well known that texturing can help trap light in solar cells. In this case, however, there are approximately 60 nm variations in height with periodicities of around 15 microns. The schematics are not to scale and greatly exaggerate the extent of the surface height variation. I wouldn't expect such a small change in the surface angle to help much. The EQE plots and the

change in J_{sc} support my position. At best, the light trapping helps just a tiny bit around 700 nm. Reducing the temperature of the diethyl ether deposition clearly had its impact by increasing V_{oc} , which is unlikely to be a result of light trapping.

(Answer) Actually, as reviewer argued, the solar spectrum absorption does not seem to be significantly improved in the wrinkled perovskite layer as shown in EQE spectrum and the simulated net absorption by rigorous coupled-wave analysis (RCWA) as shown in **Fig R1** (not included in the revised SI). There is tiny difference in the band from 750 nm. However, the wrinkle structure is on rear side of device (near back contact). Also, the wrinkle period is large ($\sim 13 \mu\text{m}$) and the amplitude is also significantly smaller than $\lambda/4$. Therefore, it is difficult to say that the wrinkled structure is particularly helpful in light management. We remove FDTD in MS to avoid misleading. As reviewer commented, our experiments show impacts to mainly increasing V_{oc} and FF by reducing T_{DE} . Also, we observed prolonged carrier lifetime as T_{DE} decreased as show in **Fig. 4**. Therefore, it is reasonable that the prolonged carrier lifetime is the main effect of the wrinkled morphology at lower T_{DE} . In literature, response of EQE spectrum at longer wavelength is significantly related to carrier collection length (L_C) which is proportional to carrier diffusion length (L_D). [Nakane et al, *J. Appl. Phys.* **2016**, 120, 064505.]

We revised MS (on p.11) as “As listed in **Table S1** and **Fig. S10**, the forward- and reverse-scanned photovoltaic measurements exhibited improvement of the overall average PCE from 19.46% (J_{sc} : 22.365 mA/cm², V_{oc} : 1.123 V, FF: 0.7729) to 21.00%. (J_{sc} : 22.662 mA/cm², V_{oc} : 1.158 V, FF: 0.8004) upon decreasing T_{DE} from 30 °C to 5 °C. In **Fig. 3b**, we can identify the possible contributions made by the wrinkled morphology. For example, in the spectral range between 600-750 nm, EQE shows enhancement for the wrinkled morphology ($T_{DE} = 5 \text{ }^\circ\text{C}$) compared to the flat reference ($T_{DE} = 30 \text{ }^\circ\text{C}$). This enhancement can be attributed by extended collection length (L_C) of carriers responsible for the longer wavelength incident photons. The extended charge carrier collection length corresponds to the longer carrier diffusion length (L_D) [23]. This implies that the enhanced photovoltaic performances of the wrinkled morphology is due mainly to the facilitated transport property of photo carriers.”

Fig. R1. Simulated net absorption by rigorous coupled-wave analysis (RCWA) of perovskite film without (flat) and with mild or enhanced wrinkle.

3. On page 16, the authors describe the FDTD simulations of how light will propagate in the solar cells. Their analysis of the data is simply “Light absorption is more enhanced in the wrinkled structure than in the flat one.” If they properly analyze their FDTD data in a quantitative way and find out the percentage change in light absorption, I think it will support what I have said in the previous paragraph.

(Answer) Regarding the question, we answered in detail. Please see the answer to the comment #2 of Reviewer 1.

4. At the end of a paper that very thoroughly examines the affect of varying the temperature of the anti-solvent and the primary solvent composition on wrinkling and optoelectronic properties, the authors suddenly say that they raised the efficiency considerably by adding some KI, which has nothing to do with the rest of the paper. They don’t actually say how much KI they added. I find that very unsatisfying. The authors must say how much additive they used. Instead of telling readers in the abstract that they reached 23 % efficiency (with KI), they should say what efficiency they obtained as a function of DE deposition temperature and explain that they get higher efficiency in the wrinkled films. That is the real take home message for this manuscript.

(Answer) We thank the reviewer for the comment. We modified abstract “A power conversion efficiency (PCE) of 21.00% is observed for the sinusoidal winkled morphology formed at $T_{DE} = 5\text{ }^{\circ}\text{C}$, which is higher than that of 19.46% for the flat one at $T_{DE} = 30\text{ }^{\circ}\text{C}$ due to the improved

voltage and fill factor.” We added information about how much KI added in “Device Fabrication” section of the revised supporting information (SI) as “For K-doped $\text{FA}_{0.92}\text{Cs}_{0.08}\text{PbBr}_{0.15}\text{I}_{2.85}$ perovskite, a precursor solution was prepared by mixing 0.1592 g of FAI, 0.4449 g of PbI_2 (99.9985%, Alfa Aesar), 0.0128 g of PbBr_2 (99.999%, Sigma Aldrich) and 0.0170 g of CsBr (99.999%, Sigma Aldrich) in 75 μL dimethyl sulfoxide (DMSO, >99.5%, Sigma Aldrich), 0.465 mL of *N,N*-dimethylformamide (DMF, 99.8% anhydrous, Sigma Aldrich) and 60 μL KI solution (0.166 g of KI (99.999%, Sigma Aldrich) in 10 mL of DMF). The precursor solution kept at temperature of 15 °C was spin-coated on the substrate (substrate temperature was 15 °C) at 4,000 rpm for 20 s, where 0.35 mL of diethyl ether ($T_{\text{DE}} = 5$ °C) was dropped in 10 s after spinning. The brownish adduct film formed right after deposition was heated at 145 °C for 10 min. After the perovskite film was cooled down to room temperature, 30 μL of 4-fluoro phenylethylammonium iodide (4F-PEAI) solution (10 mM in IPA) was spin-coated at 6,000 rpm for 20 s.” Also, we revised Fig. S17 as follows.

Fig. S17. (a) J-V curve of PSC employing the K-doped $\text{FA}_{0.92}\text{Cs}_{0.08}\text{PbBr}_{0.15}\text{I}_{2.85}$ perovskite ($T_{\text{Sub}} = 15$ °C and $T_{\text{DE}} = 5$ °C). The ratio of $[\text{Pb}^{2+}]$ to $[\text{K}^+] = 0.006$. Data were collected under A.M. 1.5G one sun illumination at scan rate of 260 mV/s. Aperture area was 0.1 cm². (b) Optical microscope image of the K-doped perovskite film.

5. I do not understand why the authors say “Since the wrinkle texture certainly has the potential for epitaxial growth due to an underlying buckling process associated with a compressive stress.” I don’t see what epitaxial growth has to do with buckling or compressive stress. The authors say this work is motivated by obtaining epitaxial growth. I only see one line on page

16 that says that an XRD plot in the supplemental section shows better orientation for the low temperature antisolvent film.

(Answer) We agreed to what the reviewer commented. We removed the term “epitaxial growth” from MS.

6. The statement on page 14 that “This indicates lower defect concentrations at the bottom side of the films, which is attributed to the crystal growth direction from the top surface to the bottom.” is not clear or well supported by data. The authors reference Stranks et al’s manuscript (ref 30) on how stress affects trap density and photoluminescence, but do not say anything about it. I recommend summarizing the main points of that manuscript and using them to try to explain the observations reported in this manuscript. Why do the authors think stress affects the trap density?

(Answer) We modified MS on p.15. “Moreover, it is also notable that k_1 is lower at the bottom than at the top (i.e., $k_1 = 3.6 \times 10^6 \text{ s}^{-1}$ in spot T-a at the top vs. $k_1 = 2.6 \times 10^6 \text{ s}^{-1}$ in spot B-f at the bottom). This indicates the bottom side of the films has the lower defect concentration. This would confirm again that that the crystal grows from the top surface (initially crystallized part with more defects) to the bottom (retarded crystallization in relatively DMSO-rich environment) which can further allow lower defects.” Also, we improved the description about the correlation between local strain, defect density and transient PL properties in the manuscript, summarizing the main points of Ref 30 now on p.14-15, reading “The difference in the recombination rate at the hill and valley sites can be attributed to the difference of the local defect densities at the hill and the valley sites. It was known that both tensile as well as compressive strain in halide perovskite thin films led to an increase in the defect density [29]. Areas with higher local strain was reported to result in faster PL decay [30]. Atomistic calculations based on the first-principle models, the defect density was indeed proportional to the degree of the local strain. Indeed, we have observed that k_1 decreases with higher amplitude wrinkle morphology formed at lower T_{DE} , which indicates that defects densities at the hill sites decreases with higher amplitude. This can be attributed to the reduced structural defects such as grain boundary defects at the hill sites because local strain is additionally alleviated at structural defects [31].”

7. In Fig 4, the authors have both a grey scale and a color scale. I don’t see how one can have two color scales for a 2D plot. We cannot see the grey scale.

(Answer) We added fluorescence count information (Gray images) at each corner of figures in **Fig. 4 a-b** in the revised MS. Figure captions were corrected accordingly.

Fig. 4. Carrier lifetime on the hills and the valleys on top and bottom of the sinusoidal wrinkled structures. Fluorescence-lifetime imaging microscopy (FLIM) images on (a) the top and (b) the bottom of the PMMA/(FAPbI₃)_{0.875}(CsPbBr₃)_{0.125}/Glass device, where the perovskite layers were prepared at $T_{DE} = 5\text{ }^{\circ}\text{C}$, $15\text{ }^{\circ}\text{C}$ and $30\text{ }^{\circ}\text{C}$ at the given $T_{Sub} = 15\text{ }^{\circ}\text{C}$. PMMA (poly(methyl methacrylate)) and glass are top and bottom side, respectively. **Insets are gray scale images.** Plots of time-correlated single-photon counting (TCSPC) measured on the

selected areas (hill (T-a, B-f, T-c and B-h), valley (T-b, B-g, T-d and B-i) and flat (T-e and B-j) on (c) the top and (d) the bottom surface, where top and bottom sites are illustrated in inset. Imaging area was $50 \times 50 \mu\text{m}^2$ and each pixel was about ~ 97 nm in diameter (excitation /PL collection spot: 512×512 pixels). (e) Plot of $\tau_{\text{av}}^{1/2}$ ($\tau_{\text{av}} = (\tau_{\text{hill}} + \tau_{\text{valley}})/2$) and photocurrent values from pc-AFM at hill and valley (3 positions of each area) in Fig. 3e of the PMMA/(FAPbI₃)_{0.875}(CsPbBr₃)_{0.125}/Glass device.

8. I don't think "Topographical tailoring" are good words to use in the title. I was partially expecting the paper to be about topological insulators, which it isn't at all. People won't know what the authors mean by "topographical tailoring." I suggest a title such as "How perovskite composition and antisolvent miscibility affect film wrinkling and optoelectronic properties."

(Answer) We changed the title as follows "How antisolvent miscibility affects perovskite film wrinkling and photovoltaic properties".

9. I think that the choice of solvents is so important that it should be stated clearly early in the main text, not just in the supplemental section. The way the manuscript is written, one might think that DMSO is the primary solvent. In fact, it is an additive and DMF is the main solvent. The authors might also want to clarify that the DMF probably evaporates first, which is probably why they write about the DMSO more.

(Answer) To clearly show that DMF is the main solvent, we modified MS on p.8 and added Figs. S4f and g on p.10 as "In particular, as shown in Fig. S4, we observed higher miscibility at relatively higher temperatures (i.e. > 25.4 °C), while phase separation of DMSO and DE mixture at relatively low temperature (i.e., < 15.2 °C) indicating less miscibility, whereas DMF, the main solvent of precursor solution, is fully miscible with DE regardless of the mixing temperature."

Fig. S4. Digital photographs of the mixture of DMSO and diethyl ether (1:8 v/v) at different solution temperatures of (a) 30.9 °C, (b) 25.4 °C, (c) 15.2 °C, and (d) 5.6 °C. (e) Photograph of the solution prepared at 5.6 °C showing that the solution was spontaneously frozen. Photographs of the mixture of DMF and diethyl ether (1:8 v/v) at different solution temperatures of (f) 29.6 °C and (g) -1.9 °C.

10. On page 4, the authors refer to “blue frames” in Fig 1. It took me a while to notice the blue frames, which are very hard to see. It is slightly hard to follow the trends in how composition affects wrinkling by looking at microscope images, many of which are too small. I really like the plots of λ vs composition in Fig S1 d and e. I recommend putting those plots in the main text. They helped me understand the trend instantly.

(Answer) According to the reviewer’s comment, we modified **Fig. 1** and sets of optical images were moved to the revised SI to clearly show trends of wrinkling depending on T_{DE} and compositions as follows.

Fig. 1. Wrinkled morphologies of perovskite thin film. (a) A schematic illustration of the experimental procedure to control wrinkled morphology in the perovskite film, together with an optical microscope image of the wrinkled morphology having amplitude (A) and wavelength (λ). Phase diagrams of the wrinkled morphology for (b) $\text{FA}_{1-x}\text{MA}_x\text{Pb}(\text{Br}_y\text{I}_{1-y})_3$ (annealed at 145 °C for 10 min) and (c) $\text{FA}_{1-z}\text{Cs}_z\text{Pb}(\text{Br}_y\text{I}_{1-y})_3$ (annealed at 145 °C for 10 min) perovskite thin films with different compositions. Color maps of wavelength (λ) and amplitude (A) are presented in the middle panels for each composition. Selected optical microscope images of a_1 - a_4 and b_1 - b_4 are presented in the right panels for each composition. (d) Optical microscope images for the $\text{FA}_{0.875}\text{Cs}_{0.125}\text{Pb}(\text{Br}_{0.125}\text{I}_{0.875})_3$ perovskite films with different temperatures of T_{DE} and T_{Sub} . Scale bar is 50 μm . Dependence of A and λ values on T_{DE} s at a given T_{Sub} is illustrated with bar graph on the right panels.

Fig. S1. Optical microscope images of the surface morphology of perovskite films with different compositions for (a) $\text{FA}_{1-x}\text{MA}_x\text{Pb}(\text{Br}_y\text{I}_{1-y})_3$, (annealed at 145 °C for 10 min) (b) $\text{FA}_{1-z}\text{Cs}_2\text{Pb}(\text{Br}_y\text{I}_{1-y})_3$ (annealed at 145 °C for 10 min) and (c) $\text{MA}_{1-w}\text{Cs}_w\text{Pb}(\text{Br}_y\text{I}_{1-y})_3$ (annealed at 100 °C for 10 min). (d) Wrinkled phase diagrams of $\text{MA}_{1-w}\text{Cs}_w\text{Pb}(\text{Br}_y\text{I}_{1-y})_3$ perovskite thin films with different compositions, together with color maps showing almost no wrinkled morphologies.

11. On page 10 (line 218), should “to form lower TDE” be “to form at lower TDE?”

(Answer) We modified the sentence.

12. Why were the substrates at 15 degrees C during spin casting? That is unusual and inconvenient. There must be a good reason.

(Answer) Since temperature near 15 °C was found to be critical in miscibility of DMSO and DE. At temperature below 15 °C, miscibility starts to decrease. Thus, to control the miscibility of DMSO/DE mixture, the critical temperature of 15 °C was applied to the substrate (see **Fig. S4** in the revised SI). In addition, temperature of substrate was varied because the substrate temperature was found to affect viscosity (η) of viscous precipitate (**Fig. R2**) and evaporation rate of solvents.

Fig. R2. Temperature dependent viscosity of an amorphous viscous precipitates obtained by pouring diethyl ether to the perovskite precursor solution for $(\text{FAPbI}_3)_{0.875}(\text{CsPbBr}_3)_{0.125}$.

Reviewer #2 (Remarks to the Author):

Park et al. reported a universal method to produce a pseudo-epitaxial perovskite layer via manipulating strain and compression relaxation in a perovskite film, which lead to a controllable wrinkled film structure. Deep study revealed the wrinkling mechanism of the compressing tensile-strained surface through a bilayer intermediate. Finally, the finding of hill sites showed longer carrier lifetimes and higher optical absorption than the valley sites, providing the basis for over 23% PCE and long-term operational stability of PSCs. This is an interesting and deep work on growth kinetics control of perovskite crystals, and the connection between topological arrangement and photocarriers behavior provides useful guidance for more efficient and stable PSCs design. However, there are still some concerns, which need more clarification:

(Answer) We appreciate the encouraging comments on our work.

(1) The wrinkling structure only occurs under some specific components (i.e. $(\text{FAPbI}_3)_{0.875}(\text{CsPbBr}_3)_{0.125}$) at given conditions (i.e. TDE is as low as 5 °C), thus it is hard to popularize this method and the cooling of diethyl ether will increase the production costs. Actually, there have been some components which may hardly form wrinkling structure (Advanced Materials, 2020, 32, 1907757; Science, 2019, 366, 749–753; Nature Photonics, 2019, 13, 460–466, etc.) but exhibit higher PCE and similar stability compared to this work,

and all of them are fabricated under simpler process. The authors should comment on this fact and the potential advantages of this method.

(Answer) It is important to tune the various parameters to maximize the PSCs performances while maintaining the number of the factors as small as possible for the practical applications. As the reviewer stated, the cooling process in manufacturing for mass production can increase the production costs. However, the method in this study does not require such a high-performance cooling process such as liquid nitrogen or huge cooling towers in petrochemical process because range of cooling temperature is moderate (0 ~ 30 °C). Therefore, influence on mass production cost may be marginal. Compared with previous studies mentioned by the reviewer, our method did not require new material and additional time for process. Our method using the miscibility difference by controlling T_{DE} is very simple. Moreover, we proposed a mechanism for forming a wrinkle structure that can encompass the entire composition of perovskite, which is quite unique as compared to previous studies. In the present study, we would like to suggest a new dimension such as morphology control to tune the photovoltaic properties of perovskite materials, which can provide additional aspect of optimizing the PV performances to the community. Actually, we are planning to extend our work to incorporate multi-scale (or hierarchical) wrinkles to address broadband solar spectrum to control further the optical path length and the photo-carrier lifetime. Based on the fundamental study and detailed data shown in the present work, researchers can obtain substantial benefit to control the thin film morphologies to improve the perovskite photovoltaic performances.

(2) The bilayer intermediate is the key for wrinkling structure formation, however, whether such an intermediate will bring into the component inhomogeneity along with the vertical direction of the perovskite films? Moreover, the existence of wrinkling structure may also lead to the component inhomogeneity between hills and valleys. The authors should provide more information on these details.

(Answer) Little change in component inhomogeneity was studied already as shown in the steady-state PL in **Fig S15** in the revised SI. No changes in peak position between hill and valley and top and bottom were observed for the wrinkled $(\text{FAPbI}_3)_{0.875}(\text{CsPbBr}_3)_{0.125}$ film (763 nm ~ 764 nm), which is consistent with the previous report (S. Braunger et al. *J. Phys. Chem. C* **2018**, 122, 17123–17135.). This reference was added as a reference [26] in the revised MS.

(3) The amplitude (A) is about 100 nm for optimized perovskite film, although this value may have no obvious influence on Spiro-OMeTAD deposition (which obtains a thickness over 200 nm), however, some more stable HTLs such as PTAA, P3HT is hard to form continuous film when depositing on the wrinkle perovskite, resulting to the existence of lots of shunting paths. This is another concern that hinders the development of this method.

(Answer) We clearly recognize the concern raised by the reviewer. Fortunately, the amplitude of the wrinkle is about 100 nm, and therefore a conformal polymeric HTL layer is expected to form onto the surface of the wrinkled perovskite film when the HTL thickness is comparable to or greater than the amplitude itself. Typically, the thickness of the polymeric HTL is around 100-200 nm, and the surface of the perovskite is favored by the polymeric HTL without concerns on the delamination or rupturing of the HTL. Therefore, the wrinkled surface of perovskite would be safe from the shunt. Also, if other deposition methods for HTL is used such as air knife, it will be freer from generation of shunting paths. [J. ding et al, *Joule*, **2019**, 3, 402–416.]

(4) In-situ absorption data showed a perovskite phase with thickness of 250 nm for TDE = 5 °C, 362 nm for TDE = 15 °C and 450 nm for TDE = 30 °C. Since anti-solvent was dropped on the top surface, it was reasonable to think that the pre-formed perovskite phase had a trend to distribute near the surface. But this evidence is not conclusive for a bilayer structure because partial pre-formed perovskite phase might also locate in bulk which could not been fully excluded so far. So, are there further characterizations or explanations on this especially a direct evidence of phase boundary?

(Answer) We appreciate the comments raised by the reviewer. We conducted additional experiment to confirm the wrinkling mechanism of a bilayer structure and modified MS and SI. We revised MS and added the modified **Fig. 2**. in p.9-10 as “To further confirm the bilayer model for the wrinkling mechanism, we have numerically simulated the morphological evolution of the thin film wrinkling based on temporal evolution of the wrinkle geometry (see details in SI) [22]. As shown in **Figs. S7** and **S8**, we can find that the bilayer model provides qualitatively similar wrinkling morphologies accompanied by 2D fast Fourier transform (FFT) images to the experimentally observed images. We have also tested again the bilayer model by examining the optical diffraction patterns of the wrinkled thin films (**Fig. 2g**). As shown in **Fig. 2h**, the optical diffraction patterns would exhibit different patterns (i.e., concentric ring patterns

for the wrinkled bilayer, while dot or single ring pattern for the wrinkled monolayer) with different configurations as denoted in **Fig. 2g**. Indeed, we observe concentric ring patterns at glass side (bottom) of film just after contacted with diethyl ether (10 s after spin started), and the patterns disappears with time, whereas the transmitted concentric ring patterns was sustained for long time as shown in **Fig. 2i**. This can be compared to the diffraction patterns of the wrinkled perovskite films obtained from reflected side and transmitted side which are commonly sustained over long time (see **Figs. S9a and b**). With the theoretical analysis supported by numerical calculations and experimental observations of the diffraction patterns, we can suggest that the wrinkling of the perovskite thin films can be elucidated by a bilayer model.”

Fig. 2. Bilayer mechanism of the **wrinkling of perovskite thin films** (a) *In-situ* photoluminescence (PL) measured in the course of spinning from 0 s to 24 s at $T_{\text{Sub}} = 15\text{ }^{\circ}\text{C}$ and $T_{\text{DE}} = 5, 15$ and $30\text{ }^{\circ}\text{C}$. (b) Plot of PL peak positions as a function of spinning time. Spin-coating duration was 20 s (dashed line). (c) *In-situ* UV-Vis absorption spectra measured during spinning from 0 s to 24 s at $T_{\text{Sub}} = 15\text{ }^{\circ}\text{C}$ and $T_{\text{DE}} = 5, 15\text{ }^{\circ}\text{C}$ and $30\text{ }^{\circ}\text{C}$. (d) The evolution of the thin solid upper perovskite layer thickness (h_f) calculated from the measured absorption data. Spin-coating duration was 20 s. (e) Viscosity of an amorphous viscous precipitates obtained by pouring diethyl ether to the perovskite precursor solution for $(\text{FAPbI}_3)_{1-z}(\text{CsPbBr}_3)_z$ with $z = 0 \sim 0.2$ and $(\text{FAPbI}_3)_{1-x}(\text{MAPbBr}_3)_x$ with $x = 0 \sim 0.6$ (see **Fig. S5**). (f) An intermediate layer

structure for the flat surface (case 1), the wrinkled structure (case 2) and the coarse surface with trace of wrinkle (case3). A bilayer model with a solid upper layer with thickness of h_f was proposed to explain the wrinkling process. (g) Schematic representation of setup for measuring optical diffraction pattern at bottom surface of bilayer film. (h) Reflected optical diffraction patterns for bilayer and monolayer films. (i) Photographs of reflected (up), transmitted (down) optical diffraction patterns as function of spin coating time (11 s, 12 s, 14 s and 16 s) after perovskite precursor is contacted with diethyl ether at 10 s ($T_{\text{sub}} = 15 \text{ }^\circ\text{C}$ and $T_{\text{DE}} = 5 \text{ }^\circ\text{C}$).

Also, we revised detail information of theoretical analysis of wrinkling of a bilayer system on p. s12-s16 in the revised SI and added **Fig. S7, S8** (with detail information) and **S9** on p. 17-19 as follows.

Theoretical Analysis of Wrinkling of a Bilayer System

1. Calculation of the governing wavelength of the wrinkle

To describe the wrinkling of a perovskite thin film observed in experiments, we can employ a simple elastic-viscoelastic bilayer model as suggested by Im and Huang [S12]. In the model, the system is modeled as an elastic layer (hereafter denoted as a subscript f)-capped viscoelastic substrate (hereafter denoted as a subscript S). Using a linear perturbation analysis, the wrinkling morphology can be approximated as a sinusoidal function with amplitude of A and period (wavelength) of λ . The development of the wrinkled morphology follows a dynamics governed by the fastest growing mode, which can be expressed with a growing constant α such that

$$\alpha \propto - \left[k^2 h_f^2 - \frac{12(1-\nu_f^2)\sigma_0}{E_f} \right], \quad k = \frac{2\pi}{\lambda}, \quad (\text{S1})$$

where h_f is the thickness, ν_f is Poisson's ratio, and E_f is elastic modulus of the elastic capping layer, respectively. The wrinkling results from the relaxation of the in-plane compressive stress denoted as σ_0 . The origin of the in-plane stress comes from the difference of mechanical responses of the elastic capping film and the underlying viscoelastic substrate. For example, we can suggest that the difference is due mainly to the thermal expansion coefficient of the two layers [S13]. The absolute value of the compressive stress developed by discrepancy of the thermal expansion coefficients can be expressed as follows.

$$|\sigma_0| = \frac{(\Delta\alpha\Delta T)E_s}{\frac{h_f}{H}(1-\nu_s) + \frac{E_s}{E_f}(1-\nu_f)}, \quad (S2)$$

where $\Delta\alpha$ denotes the difference of thermal expansion coefficients of the two layers, ν_s , E_s and H are Poisson's ratio, elastic modulus, and the thickness of the underlying viscoelastic substrate, respectively. The strain developed by the thermal expansion discrepancy is assuredly proportional to the temperature change ΔT . For most of the elastic-viscoelastic bilayer system, $\frac{E_s}{E_f} \ll 1$, and therefore, we can simplify eq (S2) as follows.

$$|\sigma_0| \approx \frac{(\Delta\alpha\Delta T)E_s H}{(1-\nu_s) h_f}. \quad (S3)$$

With the information on σ_0 in eq (S3), we can proceed to calculation of the fastest growing mode for the wrinkling amplitude from a relationship of $\left. \frac{\partial \alpha}{\partial k} \right|_{k=k_c} = 0$, $k_c = \frac{2\pi}{\lambda_c}$, where λ_c

denotes the characteristic wavelength of the wrinkle corresponding to the fastest growing amplitude, which results in

$$\lambda_c = \pi\beta h_f, \quad \beta \equiv \left(\frac{-2E_f}{3(1-\nu_f^2)\sigma_0} \right)^{1/2}. \quad (S4)$$

The critical compressive stress corresponding to can be calculated as follows [S12],

$$\sigma_c = \left(\frac{-2h_f E_f \mu_R}{3(1-2\nu_s)(1+\nu_s)H} \right)^{1/2}, \quad (S5)$$

where μ_R denotes the rubbery modulus of the underlying viscoelastic layer. In the case in which $|\sigma_0|$ is greater than $|\sigma_c|$, the bilayer system suffers morphological instability which is initiated by small fluctuation of thickness. The small fluctuation is spontaneously evolved into the wrinkle patterns with growing amplitude. From eq (S2) and (S4), we can obtain the dependence of on the thickness of the bilayer as follows

$$\lambda_c \propto h_f \left(\frac{h_f}{H} \right)^{1/2}. \quad (S6)$$

At equilibrium, typical wrinkle morphology exhibits amplitude which is considerably

smaller than period (i.e., $A/\lambda \ll 1$). For example, in our experiments, $A/\lambda \sim 10^{-2}$. Considering this fact, we can further calculate in-plane strain of the wrinkled bilayer, ε , by calculating expanded areas of the wrinkled surface relative to the flat surface such that

$$\varepsilon = \frac{\Delta l}{\lambda} = \frac{\int_0^\lambda (1 + A^2 \sin^2 kx)^{1/2} dx - \lambda}{\lambda} \approx \frac{(4A + \lambda) - \lambda}{\lambda} = \frac{4A}{\lambda}. \quad (\text{S7})$$

From eq (S3) and (S7), we can deduce an additional relationship between λ and thickness of the bilayer such that

$$\frac{A}{\lambda} \propto \frac{H}{h_f}. \quad (\text{S8})$$

Using eq (S6) and (S8), it is also possible to obtain a relationship of λ as a function of the bilayer thickness as follows:

$$A \propto H \left(\frac{h_f}{H} \right)^{1/2}. \quad (\text{S9})$$

2. Effects of antisolvent temperature on the wrinkle morphology

As shown in **Fig. 1d**, the major factor governing the wrinkle morphology of a perovskite thin film is the temperature of antisolvent diethyl ether (T_{DE}). At fixed temperature of the substrate (T_{sub}), we observed that λ increases while A decreases with increasing T_{DE} . Using the bilayer model suggested in the previous part, we can explain these dependences on T_{DE} . The role of the antisolvent is to drive solvent (i.e., DMSO) out of the spin-coated perovskite precursor film by inducing phase separation. For the sake of simplicity, let us assume that the spin-coated layer form bilayer mixture such that DMSO solution containing perovskite precursors and DE. Due to the limited miscibility of DMSO and DE, the mixture suffers phase separation, and the separation can be modeled as spinodal decomposition. For a simple binary mixture which suffers thermodynamic instability, the free energy density of the mixture, ΔF , can be modeled as a function of the composition of one of the components, ϕ [S14],

$$\Delta F = \gamma\phi(1-\phi) + k_B T (\phi \log \phi + (1-\phi) \log(1-\phi))$$

where γ denotes a constant concerning the interaction energy of two components and k_B is

the Boltzmann constant. In a typical temperature (T)-composition (ϕ) phase diagram of a binary mixture, spinodal decomposition results in two phases containing high and low compositions. In the case of DMSO solution and DE binary mixture, the molar fraction of DMSO solution in the DMSO-rich phase and the DE-rich phase can be expressed as $1-\phi_d$ and ϕ_d , respectively. Considering a fact that DMSO-solution has higher density, the separated phases form bilayer composed of upper layer containing dilute DMSO-solution and lower layer containing higher concentration of DMSO-solution. Without losing generality, for DMSO-dilute phase, the (upper) elastic film thickness would be proportional to ϕ_d , while the (lower) viscoelastic layer thickness would be proportional to $1-\phi_d$. Therefore, using eq (S6) and (S9), we can proceed to expression of λ and A as a function of ϕ_d as follows

$$\lambda_c \propto h_f \left(\frac{h_f}{1-\phi_d} \right)^{1/2} \propto (1-\phi_d)^{-1/2}, A \propto (1-\phi_d) \left(\frac{h_f}{1-\phi_d} \right)^{1/2} \propto (1-\phi_d)^{1/2}. \quad (\text{S10})$$

In eq (S10), we assumed that the thickness of the initially formed elastic layer is not dependent of ϕ_d .

Based on a typical phase diagram of spinodal decomposition, we can find that ϕ_d increases as the temperature of the binary mixture increases approaching the critical temperature. Therefore, ϕ_d for the case in which DE temperature is relatively low (i.e., $T_{DE} = 5^\circ C$) is smaller than ϕ_d for the cases of relatively high DE temperature (i.e., $T_{DE} = 15^\circ C$). Then, from eq (S10), we can find that the lower the value of T_{DE} , the lower the value of ϕ_d , and therefore, λ_c increases while A decreases. This can explain the experimental observation of the changes of λ_c and A with different T_{DE} , as reported in **Fig. 1d**.

3. Effect of the composition of perovskite materials on the wrinkling

We observed that the substitution of FA with Cs or MA and I with Br resulted in the decrease in λ and the increase in A at a certain substitution ratio. The smaller size of the substituents can increase σ_0 , which increase λ according to eq (S3) and (S4). Regarding the increased A , η is decreased with increasing the amount of Cs and Br or MA and Br (see **Fig. 1b** and **c**). According to ref S12, amplitude (A) is derived function of dimensionless growth rate (s), characteristic

time scale (τ) and formation time (t) ($A = A_0 e^{\frac{st}{\tau}}$, $s = \alpha - \mu_R/E_f$ and $\tau = \eta/E_f$). The A is exponentially anti-proportional to η . Therefore, when η is decreased, A is enlarged. Except for the specific ratio, however, the compositions with $z \geq 0.25$ or $x \geq 0.8$ formed a solid bottom layer, which leads to a very large η (see **Fig. S6**) and thereby a significant increase of characteristic time scale (τ) to about $10^4 \sim 10^5$ times, resulting in less formation of wrinkled texture.

4. Effect of the annealing condition on the wavelength of the wrinkles

Given a condition of $\sigma_0 > \sigma_c$, wrinkling starts with long wavelength (λ_0) which will be eventually narrowed and saturated as the stress is being relaxed until $\sigma_0 = \sigma_c$ [S15]. However, in perovskite film formation process, λ_0 cannot be saturated because the bottom layer is solidified before it is saturated, which may lead to a residual compressed stress after spin-coating [S16]. The slight decrement of λ after annealing is evidence of the presence of residual stress because the relaxation of residual stress will further decrease λ as shown in **Figs. S2d** and **S2e**.

5. Null contribution of E_f and v_f

E_f and v_f can be also assumed to be constant due to a small difference in E_f between 10.2~11.8 GPa for FAPbI₃ and 9.7~12.3 GPa for FAPbBr₃ even upon replacing iodide with bromide [S17] and small v_f of perovskite (0.28~0.33) [S18].

6. Effect of T_{Sub}

At fixed temperature such as $T_{DE} = 15^\circ\text{C}$, λ increases, while A decreases with increasing T_{Sub} from 5°C to 15°C (see **Fig. 1d**). Upon increasing T_{Sub} , h_f is expected to increase because the miscibility between DMSO and diethyl ether is enhanced by elevating T_{Sub} . This can lead to an increase in h_f but decreases in A .

Evolution of the wrinkle pattern of the bilayer

To confirm the wrinkling mechanism of a bilayer structure observed in our experiments, we provide a computer simulation of the temporal morphological evolution of the surface wrinkles of the bilayer. For this simulation, we employed a typical finite-difference method for 2D simulation box (800×800) with periodic boundary condition. According to the theoretical and numerical scheme suggested by Im and Huang [S12], we modeled the morphological evolution

of the bilayer wrinkles as shown in **Fig. S7**. As shown in **Fig. S8**, one can find that the simulated wrinkle morphology is similar to the experimentally observed morphology. The similarity is confirmed again by comparing the 2D fast Fourier transform (2D FFT) signals obtained from the simulated and experimentally observed morphologies, in which isotropic wrinkles pattern with notable concentric ring patterns which correspond to the characteristic length scale (i.e., λ_C) of the wrinkles. The computer simulated bilayer wrinkle morphology strongly supports that the wrinkling mechanism of the perovskite thin film hinges on the relaxation of the in-plane compressive stress developed in the elastic-viscoelastic bilayer.

Fig. S7. Numerical simulation results of the morphological evolution of perovskite (PSK) wrinkles of elastic-viscoelastic bilayer. τ denotes dimensionless time scale. The left column is for the time-dependent surface wrinkle morphology. The inset for 1000τ shows a 5 times magnified image. The middle column is the 2D FFT patterns of the wrinkles. The right column is the cross-sectional profiles of the wrinkles along the horizontal (namely x-cross) and vertical directions (namely y-cross).

Fig. S8. Comparison of the morphologies of the wrinkle pattern obtained from (a) numerical calculation and (b) experimental observation. 2D FFT patterns for (c) numerically calculated wrinkles and (d) experimentally observed wrinkles (c for a and d for b).

Fig. S9. (a) Schematic representation of setup for measuring optical diffraction pattern at top surface of bilayer perovskite film. (b) Photographs of the reflected (up) and the transmitted (down) optical diffraction patterns at top surface as function of spin coating time (11 s, 12 s, 14 s and 16 s) after perovskite precursor was contacted with diethyl ether at 10 s. $T_{\text{Sub}} = 15\text{ }^{\circ}\text{C}$ and $T_{\text{DE}} = 5\text{ }^{\circ}\text{C}$. (c) UV-Vis spectrum of glass substrate and viscous precipitate obtained by pouring diethyl ether to the $(\text{FAPbI}_3)_{0.875}(\text{CsPbBr}_3)_{0.125}$ perovskite precursor solution.

(5) How to understand the decreased defect densities in the wrinkle structure compared with the flat morphology? Moreover, why the hill sites show longer carrier lifetimes than valley sites? The authors should give more description on the deep mechanism.

(Answer) First of all, the wrinkle structure results from the stress relaxation across the thin film, and therefore, it is more probable to have lower local strain-concentrated geometry which can host local defects like fractures, grain boundary defects compared to the case of a flat thin film. In the revised MS, we now address this point by adding the following sentences on p. 14 “From the exponential fitting of the PL decay curve, we can deduce the decaying rate k_1 such that $PL(t) \propto \exp(-2 \cdot k_1 t)$ (see the **SI** for details and **Table S2**). From the exponential fitting, we found that k_1 is lower at the hill sites, while higher at the valley sites. Also, it decreases as T_{DE} is lowered (i.e., $3.6 \times 10^6 \text{ s}^{-1}$ or $4.4 \times 10^6 \text{ s}^{-1}$ for the spots T-a or T-c (hill sites) vs. $5.7 \times 10^6 \text{ s}^{-1}$ or $7.4 \times 10^6 \text{ s}^{-1}$ for the spots T-b or T-d (valley sites)). The difference in the recombination rate at the hill and valley sites can be attributed by the difference of the local defect densities at the hill and the valley sites. It was reported that both tensile as well as compressive strain in halide perovskite thin films lead to an increase in the defect density [29]. Areas with higher local strain can result in faster PL decay [30]. Atomistic calculations based on the first-principle models, the defect density is indeed proportional to the degree of the local strain. Indeed, k_1 is observed to decrease with higher amplitude wrinkle morphology (with lower T_{DE}), which indicates that defects densities at the hill sites decrease with amplitude. This can be attributed to the reduced structural defects such as grain boundary defects at the hill sites because local strain is additionally alleviated at structural defects [30]. The reduced grain boundary defects should be accompanied by the enhanced uniformity of the grain sizes, which can be checked by the narrower distribution of the grain areas (see **Figs. S16**.)” and we added Fig S16 in the revised SI.

Fig. S16. Distributions of grain area (μm^2), extracted from the images in **Fig. S13**, of (a) hill and (b) valley for $T_{DE} = 5\text{ }^\circ\text{C}$, (c) hill and (d) valley for $T_{DE} = 15\text{ }^\circ\text{C}$ and (e) flat for $T_{DE} = 30\text{ }^\circ\text{C}$. T_{Sub} was $30\text{ }^\circ\text{C}$. (f) Standard deviation and normalized distribution (inset) of grain size depending on T_{DES} .

(6) A minor question is that some calculation details can be put into SI, and the description of the connection between perovskite components and wrinkling structures can be more concise so that the readers can easily get the important information.

(Answer) We moved some calculation details to SI for the readers.

Reviewer #3 (Remarks to the Author):

This work by S.-G. Kim et al reports a systematic study on the wrinkle-like texture formation in a series of halide perovskite thin films. The structure and morphology of the wavy texture was controlled by varying the anti-solvent and substrate temperatures. The formation mechanism was investigated using a series of characterization tools such as in-situ PL and XRD. Using these textured perovskite thin film, solar cells with high efficiency up to 23% was fabricated. Although the formation of the textured surface has been reported by a number of

groups previously, this work is more comprehensive and contains some interesting results. However, I feel that there are large amounts of inconsistency and many of the claims in this paper can not be supported by the experimental data. My specific points are listed below. Because of this, I do not recommend publication of this work, at least in the present form.

(Answer) We thank the reviewer for the positive response to our work.

1. The authors mentioned epitaxial growth several times, however, there is no epitaxial growth in this work.

(Answer) We agree with the reviewer's opinion. We removed all expressions of epitaxial growth from MS.

2. Regarding the growth mechanism:

a. I can not understand where the lattice strain comes from as the growth from the TiO₂ layer is not epitaxial.

b. Then it is also hard to understand why the "wrinkle" surface can release the strain as the wrinkle is very much macroscopic with a wavelength on the order of micrometers.

(Answer) For both questions of (a) and (b), wrinkling (not epitaxial, we eliminated "epitaxial") is not related to the substrate ETL material morphology but related to a bilayer model upon antisolvent treatment in our work. The lattice strain is also related to and strongly dependent on temperature of diethyl ether. From the bilayer mechanism in perovskite film formation, the rapidly generated thick upper elastic layer (i.e. T_{DE}=30 °C) hinders relaxation of compressive stress at surface, leading to a flat surface, while thin h_f can release most of compressive stress at surface by forming wrinkled surface before crystallization. Therefore, flat perovskite shows irregular grain shape and many cracked or embedded grains in SEM image. We revised MS on p.14-15 as "From the exponential fitting of the PL decay curve, we can deduce the decaying rate k_1 such that $PL(t) \propto \exp(-2 \cdot k_1 t)$ (see the S.I. for details and Table S2). From the exponential fitting, we found that k_1 is lower at the hill sites, while higher at the valley sites. Also, it decreased as T_{DE} lowered (i.e., $3.6 \times 10^6 \text{ s}^{-1}$ or $4.4 \times 10^6 \text{ s}^{-1}$ for the spots T-a or T-c (hill sites) vs. $5.7 \times 10^6 \text{ s}^{-1}$ or $7.4 \times 10^6 \text{ s}^{-1}$ for the spots T-b or T-d (valley sites)). The difference in the recombination rate at the hill and valley sites can be attributed by the difference of the local

defect densities at the hill and the valley sites. It is known that both tensile as well as compressive strain in halide perovskite thin films lead to an increase in the defect density. [29] Areas with higher local strain can result in faster PL decay [30]. Atomistic calculations based on the first-principle models, the defect density is indeed proportional to the degree of the local strain. Indeed, we observed that k_1 decreases with higher amplitude wrinkle morphology (with lower T_{DE}), which indicates that defects densities at the hill sites decreases with higher amplitude. This can be attributed by the reduced structural defects such as grain boundary defects at the hill sites because local strain is additionally alleviated at structural defects.[30] The reduced grain boundary defects should be accompanied by the enhanced uniformity of the grain sizes, which can be checked by the narrower distribution of the grain areas (see **Figs. S16**).” and we added Fig S16 in SI.

c. Furthermore, Fig 3a and 3b seems no difference to me. The peaks at low angles are so broad. It is unclear to me how the peak values were selected and how Fig 3c was generated.

(Answer) It might be our instrumental broadening from divergence of beam is reason of broaden peak at low incident angle. Peak values were selected from fitting with Gaussian distribution function. As pointed out by the reviewer, GIXRD with small shift looks like almost the same peaks. The difference between them is only about 0.02 degrees even at a low incidence angle such as 2 degrees. Given the precision of XRD, it can be included in the measurement error range. Thus we removed GIXRD data from MS.

d. The in-situ PL does not provide constructive information regarding the formation mechanism.

(Answer) We disagree with the Reviewer. Based on the in-situ PL, we are able to observe a confinement effect at the beginning of film formation, which manifests itself in a broadened PL peak at shorter wavelength compared to the PL of the final film. In the course of spin-coating, the PL as well as the band edge continue to shift to longer wavelengths. This could be caused by a decreasing confinement effect or by a change in stoichiometry. Since the PL peak width remains constant, we can exclude a decreasing confinement effect as cause for the spectral shift. This finally allows to associate the spectral shifts in PL and absorption with a change in stoichiometry during film formation. Thus, our PL analysis provides fundamental insights and constructive information regarding the film formation. Details about the analysis

approach of the in-situ PL and absorption measurements are described in the SI. To better emphasize the conclusions that can be drawn from the PL spectra, we rephrased the corresponding section in the SI on p. s9: “The slower shift for longer time could either stem from a change in stoichiometry or a decreasing confinement effect. Since the PL peak width remains constant during this shift, we can exclude a decreasing confinement effect as cause for this spectral shift, so that we associate it with a change in stoichiometry during spin coating (not annealed film). This indicates a preferential formation of a bromine-rich phase, possibly e.g. due to differences in the enthalpy of formation, as reported for MAPbI₃ and MAPbBr₃ [S10], or different diffusivity in solution of the different compounds”

e. There is no experimental evidence for the bi-layer model.

Answer) We explained with experimental evidence. Please see **Answer for Comment #4 of Reviewer 2**.

3. Regarding Fig 4: It can be seen that in general the lower diethyl ether dripping temperature lead to longer lifetime for the whole film, and the lifetime for the bottom side of the film is better than top, but I do not see the "hill" is better than the "valley".

(Answer) FLIM and TCSPC were measured by pixel-wise (512×512 pixels) after pre-focusing on surface as shown in **Fig. R4**. For scanning top surface, step height between hill and valley (~240 nm) is spatially resolved by scanning focal plane with each 20 nm intervals. For scanning bottom surface, the actual surface is flat because it is attached to the glass surface. Therefore, hill and valley is confirmed by comparing optical microscope image with FILM data. A glass substrate (SiO₂) and PMMA are inert to light source and PL from samples. From **Fig. 4c** and **d**, it can be seen that at the hills the PL decays slower, i.e. the extracted k_1 value is lower compared to the valley. As discussed in MS on p.14, k_1 correlates with the defect density. From this we can conclude that the hills have lower defect density, which also leads to an increase of photocurrent in the conductive AFM measurements (as described on p.12-13), demonstrating their superior optoelectronic properties compared to the valleys.

Fig R3. Schematic representation of fluorescence lifetime imaging microscopy (FLIM) coupled with time-correlated single-photon counting (TCSPC) setup. (a) optical microscope part and (b) detector part.

4. Line 320: the authors claim that the bottom of the film has lower defect density because the crystals growth from the top to bottom. I cannot understand why growth from top to bottom will lead to lower defect density at the bottom. More explanation and clarification are necessary.

(Answer) The top part of the film in the course of the crystallization would support subsequent crystallization downward the bottom. In particular, considering a fact that there is no significant composition variation along the thickness direction of the film, the crystallization is basically homogeneous crystallization, which can further allow lower defects compared to the initially crystallized part (Top). We revised MS on p.14 as “it is also notable that k_1 is lower at the bottom than at the top (i.e., $k_1 = 3.6 \times 10^6 \text{ s}^{-1}$ in spot T-a at the top vs. $k_1 = 2.6 \times 10^6 \text{ s}^{-1}$ in spot B-f at the bottom). This indicates the bottom side of the films has the lower defect concentrations. This would confirm again that that the crystal grows from the top surface (initially crystallized part with more defects) to the bottom (retarded crystallization in relatively DMSO-rich environment) which can further allow lower defects.”

5. Line 349: regarding the conducting AFM, the higher photoresponse may be from the larger thickness (more light absorption) at the "hill" sites over the "valley" sites. To me, this data is not convincing enough to prove the hill is better than the valley.

(Answer) In our research, we use weak light intensity (0.228 mW/cm^2 , $1 \text{ Sun} = 100 \text{ mW/cm}^2$) for pc-AFM. So, our pc-AFM results could show more surface sensitive property. Also, the

carrier concentration formed by light is higher near the surface to which light is incident, not uniform across the thickness of perovskite film. In literature, concentration of photo-generated carriers is reduced to about 1/10 at 412 nm and to 1/100 at 709 nm compared to near surface (~67 nm). [Adv. Energy Mater. 2020, 10, 1903653] Therefore, it is expected the difference in photo current between PSK film with 460 nm and 700 nm is less than 4%. However, the perovskite layers in our research have 460 nm to 700 nm (Winkled) and 550 nm (Flat) which are well applied for the above explanation, whereas pc-AFM shows significant variation of 62.4 pA to 88.6 pA (max. 94 pA) for Winkled perovskite (Variation: ~33%). To check dependence of perovskite thickness on the measured photocurrent at pc-AFM, we conducted additional experiments with the flat $(\text{FAPbI}_3)_{0.875}(\text{CsPbBr}_3)_{0.125}$ film for the different film thickness of 477 nm and 693. The result shows that difference of thickness has only a small dependency (~4-5%) which cannot explain enhancement of photocurrent in our results. We revised MS on p.12 and added **Fig. S12** in the revised SI, as follows. “To study the dependence of perovskite thickness on the measured photocurrent, J-V curves and pc-AFM are measured for the flat $(\text{FAPbI}_3)_{0.875}(\text{CsPbBr}_3)_{0.125}$ films with different film thickness of 477 nm and 693 nm. As shown in **Fig. S12**, the difference in the measured J_{sc} and in the pc-AFM photocurrent is negligible, i.e. $\Delta J_{sc} = 0.5 \text{ mA/cm}^2$ and $\Delta \text{photocurrent} = \sim 2.9 \text{ pA}$, respectively. This indicates that the photocurrent difference at the hill and valley sites is not due to the thickness difference. In addition, it was reported that the concentration of photo-generated carriers is reduced to about 1/10 at a depth of 412 nm under the film surface and to 1/100 at 709 nm as compared to the carrier concentration at near surface (~67 nm) [24]. Therefore, it is expected that difference in photocurrent between 477 nm- and 693 nm-thick perovskite layers is less than 4%. These findings allow us to explore another mechanism for the photocurrent enhancement given by the micro wrinkle morphology. (i.e., ~ 470 nm (Valley, $T_{DE}=5 \text{ }^\circ\text{C}$) and ~700 nm (Hill, $T_{DE}=5 \text{ }^\circ\text{C}$)).”

Fig. S12. (a) J-V curves of PSCs (FTO/SnO₂/Perovskite/Spiro-MeOTAD/Au) employing the flat (FAPbI₃)_{0.875}(CsPbBr₃)_{0.125} films, formed at T_{DE} = 30 °C, with different thickness of 693 nm and 477 nm. (b) Photocurrent profiles of the flat (FAPbI₃)_{0.875}(CsPbBr₃)_{0.125} films (FTO/SnO₂/Perovskite) depending on film thickness. Inset is height profile of (FAPbI₃)_{0.875}(CsPbBr₃)_{0.125} films.

Therefore, we think that higher photocurrent signal at the hill area comes from longer carrier lifetime (τ) as shown in **Fig 4** and diffusion length (d) which is proportional to $\tau^{0.5}$ as shown in the revised **Fig.4e**. So, we revised MS on p.15 as follows “This can be confirmed again by the comparison of the photocurrent and average carrier lifetime as shown in **Fig. 4e**. As compared in the plot, the average diffusion length of the photo carriers which is proportional to the square root of the lifetime of top and bottom surface, $\tau_{av}^{1/2}$, is strongly correlated with photocurrent. Considering a fact that the lifetime is inversely proportional to defect density (N_d), we can deduce that the hill sites with higher amplitude is clearly associated with lower defects than flat or valley sites.” **Fig. 4e** was included in the revised MS.

6. Based on the optical simulation, the light absorption is greatly enhanced in the textured film, but there is almost no difference in the device's photocurrent density.

(Answer) Regarding the reviewer's concern, please see the answer to the comment #2 of Reviewer 1. Since there is no significant enhancement of J_{SC}, the optical simulation part is eliminated from MS. We modified sentence on p. 11 in the revised MS as follows. “More specifically, the amplitude of the wrinkles (~100 nm) is sufficiently smaller than the quarter of the wavelength of visible and near infrared light. In addition, the spatial periodicity such as wavelength of the wrinkles is sufficiently greater than the wavelength of visible and near infrared incident light, which limits the grating effects of the micro-structures”.

7. Finally, the authors did not provide experimental details of the FA0.92Cs0.08PbBr0.15I2.85 solar cell fabrication in the method part. It is also confusing why the authors study the FA0.875Cs0.125 film in Figure 4 and part of Figure 5 and then switch to FA0.92Cs0.08 for solar cell fabrication.

(Answer) Regarding the reviewer's question, we already answered. Please see the answer to the comment #4 of Reviewer 1.

REVIEWERS' COMMENTS

Reviewer #1 (Remarks to the Author):

I'm satisfied with the revisions and think that the authors have made enough progress on an important subject to warrant publication Nature Communications.

Reviewer #2 (Remarks to the Author):

The authors have addressed most of my previous concern, and therefore I recommend its publication in Nature Communications.

Reviewer #3 (Remarks to the Author):

My questions have been addressed and the paper can be accepted in my opinion.